# Strategic Planning: A Top-Down Approach to Option Generation

Max Ruiz Luyten [* 1]   Antonin Berthon [* 1]   Mihaela van der Schaar [1]

## Abstract

Real-world human decision making often relies on strategic planning, where *high-level* goals guide the formulation of sub-goals and subsequent actions, as evidenced by domains such as healthcare, business, and urban policy. Despite notable successes in controlled settings, conventional reinforcement learning (RL) follows a *bottom-up* framework, which can struggle to adapt to real-world complexities such as sparse rewards and limited exploration budgets. While methods like hierarchical RL and environment shaping provide partial solutions, they frequently rely on either ad hoc designs (e.g. choosing the set of high-level actions) or purely data-driven discovery of high-level actions that still requires significant exploration. In this paper, we introduce a *top-down* RL framework that explicitly leverages *human-inspired strategy* to reduce sample complexity, guide exploration, and enable high-level decision making. We first formalize the *Strategy Problem*, which frames policy generation as finding distributions over policies that balance *specificity* and *value*. Building on this definition, we propose the *Strategist* agent—an iterative framework that leverages large language models to synthesize domain knowledge into a structured representation of actionable strategies and sub-goals. We further develop a *reward shaping methodology* that translates these strategies expressed in natural language into quantitative feedback for RL methods. Empirically, we demonstrate that our framework significantly enhances the performance of different underlying RL algorithms, leading to faster convergence and the discovery of more complex behaviors. Taken together, our findings highlight that *top-down strategic exploration* opens new avenues to improve RL in real-world decision problems.

[*]Equal contribution [1]University of Cambridge. Correspondence to: Max Ruiz Luyten <mr971@cam.ac.uk>.

*Proceedings of the 42nd International Conference on Machine Learning*, Vancouver, Canada. PMLR 267, 2025. Copyright 2025 by the author(s).

## 1. Introduction

**Real-world complexity and top-down planning.** Real-world human decision-making hinges on strategic goal-setting and top-down planning (Correa et al., 2023; Collins et al., 2024). From large-scale business decisions to complex healthcare protocols (Strickland & Lizotte, 2021), experts frequently define high-level objectives and then decompose them into increasingly concrete subgoals to guide daily actions. This contrasts sharply with the way conventional reinforcement learning (RL) often proceeds: *bottom-up*, through extensive trial-and-error in controlled or simulated environments (Mnih et al., 2015; Silver et al., 2016). Although bottom-up RL has led to landmark successes in various domains, its reliance on massive exploration and repeated resets conflicts with real-world domains, where errors can be costly or time-consuming (Dulac-Arnold et al., 2021), and rewards are sparse and delayed (Rengarajan et al., 2022).

**Motivation: human-inspired strategies vs. bottom-up RL.** Top-down planning is standard for humans operating in complex, uncertain, and resource-constrained contexts (Correa et al., 2023). In business, strategists consider alternative pathways for growth—such as increasing sales volume, improving operational efficiency, or diversifying product lines—before committing to one. In healthcare, medical teams identify key bottlenecks—e.g., availability of caregivers or patient discharge protocols—and define action plans that interleave resource allocation and clinical procedures. Such *human-inspired* strategy formation not only structures the planning process, but also reduces the exploration burden by focusing attention on the most promising areas of the decision space (Sutton et al., 1999; Collins et al., 2024).

**Challenges in real-world exploration.** In many real-world contexts, an unstructured bottom-up search may be neither feasible nor safe (Dulac-Arnold et al., 2021). If a learning agent suffers significant penalties (like budget overruns in a store or catastrophic patient care failures in a hospital), it may never gather enough experience to converge on a truly optimal policy. Even when exploration is feasible, the sheer scale of real-world decision spaces and the unpredictability of evolving environments can overwhelm trial-and-error learning.

---

### Illustrative Example: Collecting Meat

**Environment: Minecraft** 🟫
**Goal: Maximize Meat Collection** 🥩

#### Bottom-up Approach

- Learn low-level actions like movement.
- Trial-and-error explore to locate animals.
- Eventually discover hunting for meat.

#### Top-down Approach

- **Identify overarching strategies**: hunting, farming, or trading.
- **Outline sub-goals**: crafting tools, building enclosures, breeding livestock.
- **Plan transitions**: switch to farming once stable infrastructure is in place.

*Summary:* Bottom-up RL gets drawn to whichever approach yields immediate rewards—hunting. A top-down strategy broadens the horizon to better, more complex, approaches like farming.

---

**Recent progress and key limitations.** Recent progress in *hierarchical RL* (Kulkarni et al., 2016; Liu et al., 2024a), *environment shaping* (Park et al., 2024), and *LLM-assisted RL* (Huang et al., 2022; Ahn et al., 2022; Singh et al., 2022; Pternea et al., 2024; Yan et al., 2024) offers partial answers but still leaves key gaps. Many current methods rely on manually engineered sub-goals, or a purely data-driven discovery of structure. Moreover, most approaches often generate a single plan rather than exploring multiple promising strategies. To address these limitations, we propose a new *top-down* RL framework that uses systematic *strategic reasoning* to reduce sample complexity, guide exploration, and elevate decision-making to more abstract planning layers.

**Contributions.**

- We **formalize strategy** as a distribution over policies that balances *specificity* (pruning the search space) and *value* (encompassing optimal plans).

- We **instantiate** these ideas in the **Strategist** agent[1], which uses the LLM-based tree search to encode domain knowledge into *actionable* top-down strategies without prespecifying their components. Crucially, we introduce a *reward shaping methodology* that translates strategies into quantitative feedback for RL.

- We **empirically validate** that this procedure enables effective exploration in different task specifications in the Crafter environment (Hafner, 2021), leading to **faster convergence** and **improved final performance** when

---

[1] https://github.com/antoninbrthn/strategist

paired with PPO (Schulman et al., 2017) or SOTA methods such as EDE (Jiang et al., 2023).

We begin by formally introducing a novel top-down RL framework named the *Strategy Problem*, along with theoretical grounding for its practical efficacy in § 2. Then we discuss the existing work and how they differ from our *top-down* approach in § 3. § 4 presents a first instantiation of this framework in the *Strategist Agent*, combining the LLM-based tree search with state-based reward shaping to train strategy-guided RL agents. We demonstrate the efficacy of this method compared to traditional RL methods in § 5, and we discuss potential future directions in § 6.

## 2. The Strategy Problem

**Informal Motivation: Why a Strategist?** Real-world RL problems often involve vast state and action spaces, sparse rewards, and limited exploration budgets (Dulac-Arnold et al., 2021). In such settings, standard "bottom-up" RL (e.g., Q-learning (Watkins & Dayan, 1992)) without prior knowledge requires a prohibitively large number of trials, on the order of $O\left(\frac{|S||A|}{\epsilon^2(1-\gamma)^2}\log\frac{1}{\delta}\right)$ to provide an $\epsilon$-optimal policy with probability $1-\delta$ (Strehl et al., 2009; Lattimore & Hutter, 2012). In contrast, humans tackle complexity by proposing *multiple* plausible high-level strategies—each narrowing the search in different ways—and iteratively refining them until one is chosen. Motivated by this, we define a *Strategist* as an agent that constructs *distributions* over policies that incorporate expert inductive bias to reduce the search space while encompassing near-optimal policies.

### 2.1. Formalism Context and Definitions

**RL in a Nutshell** Let $\mathcal{M} = (\mathcal{S}, \mathcal{A}, P, R, \gamma)$ denote a Markov Decision Process (MDP), where $\mathcal{S}$ is the state space, $\mathcal{A}$ the action space, $P(s' \mid s, a)$ the transition function, $R(s, a)$ the reward function, and $\gamma \in [0, 1]$ the discount factor. A policy $\pi$ maps states to actions (or action distributions), and its expected discounted return is $J(\pi) = \mathbb{E}_{s_0 \sim \rho}[V^\pi(s_0)]$, where $V^\pi(s) = \mathbb{E}_\pi\left[\sum_k \gamma^k R(s_k, a_k)\middle| s_0 = s\right]$ is the value function and $\rho$ the initial state distribution. RL provides algorithms $\mathscr{A} : \mathcal{M} \to \Pi_\mathcal{M}$ that aim for an optimal policy $\pi^\star \in \arg\max_{\pi \in \Pi_\mathcal{M}} J(\pi)$, with $\Pi_\mathcal{M}$ as the set of policies.

**Distributions over Policies** Rather than searching over $\Pi_\mathcal{M}$ from scratch, we propose the use of a *strategist*:

$$\mathscr{S} : \mathcal{M} \times \mathcal{K} \longrightarrow \Delta(\Pi_\mathcal{M}),$$

where $\mathcal{K}$ encapsulates high-level or "common sense" knowledge about the domain (e.g., a description of the environment in natural language). The output $\delta_\Pi = \mathscr{S}(\mathcal{M}, \mathcal{K})$ is a probability distribution over policies. Intuitively, $\delta_\Pi$ restricts the policy space to "plausible" strategies in an option-

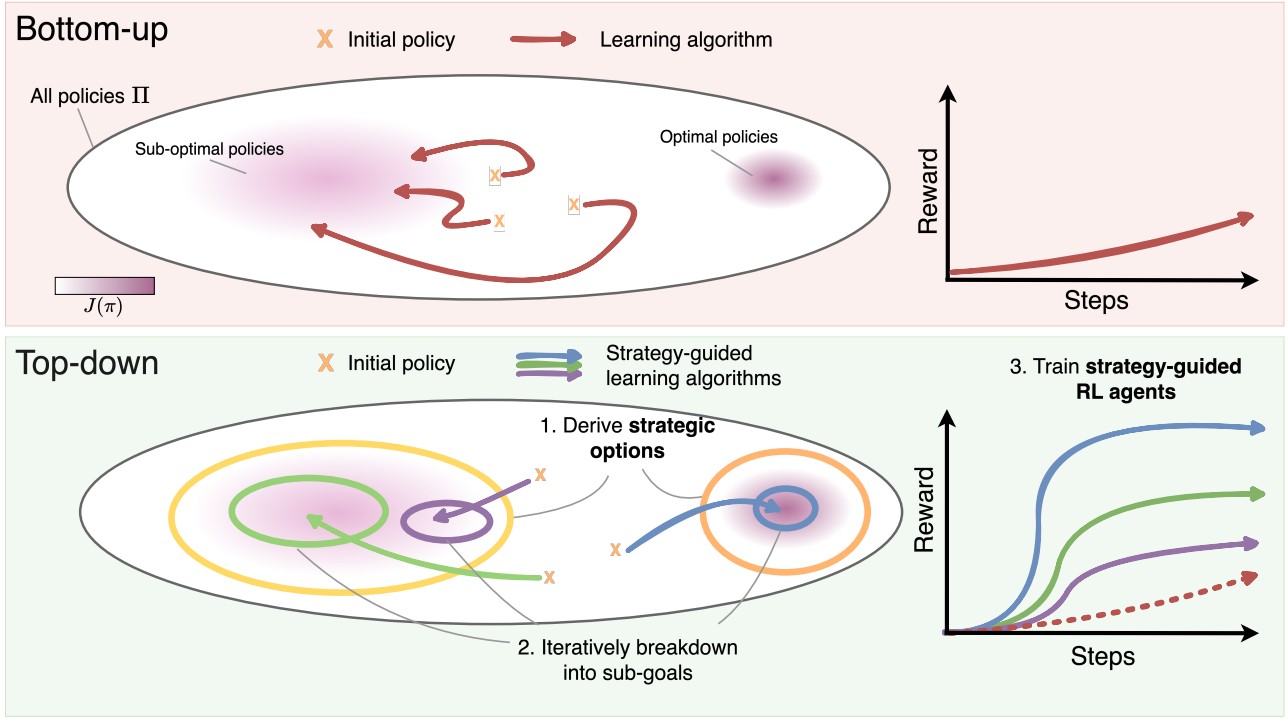

Figure 1. Comparison between bottom-up and top-down frameworks when searching for optimal policies given a goal to maximize the expected returns $J(\pi)$.

spirited manner (Precup et al., 1998), focusing learning or planning on a smaller and more promising subset.

**Specificity and Value**   A good strategy $\delta_\Pi$ must balance two key properties:

1. **Specificity**: $\delta_\Pi$ should focus on a relatively small set of promising policies, thus reducing the effective search space. We quantify the specificity through the entropy $H(\delta_\Pi)$. A "sufficiently specific" strategy satisfies

$$H(\delta_\Pi) \leq H_{\max},$$

   for some small threshold $H_{\max}$.

2. **Value**: Among the policies in $\delta_\Pi$'s support, there should exist at least one near-optimal policy. Formally, define the $\varepsilon$-support of $\delta_\Pi$ as

$$\mathrm{supp}_\varepsilon(\delta_\Pi) = \left\{ \pi \in \Pi_{\mathcal{M}} \,\middle|\, f_{\delta_\Pi}(\pi) > \varepsilon \right\},$$

   where $f_{\delta_\Pi}(\pi)$ is the density of $\delta_\Pi$. Let $\pi_{\delta_\Pi}^{*\varepsilon} = \arg\max_{\pi \in \mathrm{supp}_\varepsilon(\delta_\Pi)} J(\pi)$ be the best policy within that support. The strategy $\delta_\Pi$ is said to have *value* at least $1 - v_{\min}$ if

$$J(\pi_{\delta_\Pi}^{*\varepsilon}) \geq (1 - v_{\min})J(\pi^\star),$$

   where $v_{\min} > 0$ is a small constant.

### 2.2. Formal Definition of the Strategy Problem

> **The Strategy Problem**
>
> Given an MDP $\mathcal{M}$, high-level knowledge $\mathcal{K}$, and thresholds $(H_{\max}, \varepsilon, v_{\min})$, find a distribution
>
> $$\delta_\Pi \in \Delta(\Pi_{\mathcal{M}})$$
>
> such that:
>
> 1. **Specificity:** $H(\delta_\Pi) \leq H_{\max}$.
>
> 2. **Value:** $J(\pi_{\delta_\Pi}^{*\varepsilon}) \geq (1 - v_{\min})J(\pi^\star)$.

In essence, the Strategy Problem searches for one or multiple distributions over policies $\delta_\Pi$ that maintain low entropy (i.e. high specificity) while ensuring that its support contains at least one high-value policy.

### 2.3. How Specificity and Value Aid Sample Efficiency

The Strategy Problem aims to improve sample efficiency by constraining the policy search to a structured distribution $\delta_\Pi$ that balances specificity and value. In traditional RL, standard probably approximately correct (PAC) arguments show that finding a near-optimal policy requires a number of samples scaling with $\mathcal{O}(|S||A|)$ (Strehl et al., 2009), making

learning in large state-action spaces infeasible. In contrast, a well-designed $\delta_\Pi$ focuses exploration on a smaller, high-potential subset of policies, reducing the effective search space in an options-like spirit (Precup et al., 1998; Fruit & Lazaric, 2017) while still ensuring near-optimality. The following theorem formalizes this advantage, showing that the sample complexity depends on the entropy of $\delta_\Pi$ rather than the full size of the policy space.

**Theorem 2.1** (High Specificity & Value $\implies$ Sample Efficiency). *Consider a strategy $\delta_\Pi \in \Delta\Pi_{\mathcal{M}}$ with high specificity, i.e. $H(\delta_\Pi) \leq H_{\max}$, and high-value, so that $J(\pi^\star) - J(\pi^{*\tau}_{\delta_\Pi}) \leq \varepsilon/2$. Then, with probability at least $1 - \delta$, identifying a policy whose return is $\varepsilon$-close to optimal requires at most*

$$\mathcal{O}\left( \frac{H_{\max}}{\tau \ln\left(\frac{1}{\tau}\right)} \cdot \frac{1}{\varepsilon^2} \cdot \ln\left( \frac{H_{\max}}{\delta \tau \ln\left(\frac{1}{\tau}\right)} \right) \right)$$

*episodes, where $\tau$ determines the lower probability threshold for the candidate policies.*

The proof is provided in § A. This theorem underscores the rationale behind the Strategy Problem: the more concentrated $\delta_\Pi$ is (i.e., the lower its entropy), the fewer candidate policies, simplifying the search for a near-optimal policy. Specificity ensures a reduced search space, while the value condition guarantees that there is at least one near-optimal policy within it. As a result, sample complexity depends on $H_{\max}$ rather than the full state-action space, reinforcing the key idea that structured policy distributions can significantly accelerate RL by narrowing the search space while preserving near-optimality. Although this theoretical bound does not extend to our implementation, it provides intuition for the properties that we should strive for in a strategy.

## 3. Related work

Integrating high-level knowledge and structure into RL has long been studied under various guises of hierarchical policies, curriculum learning, and, most recently, LLM-based approaches. While these directions offer partial remedies to the challenges of sparse rewards and massive exploration costs, they often adopt a *bottom-up* perspective that requires either extensive trial-and-error or ad-hoc high-level designs. Our work instead proposes a *top-down* framework that systematically leverages human-inspired strategy formation.

**LLM-as-policy** A number of recent methods incorporate LLMs to guide RL policies by providing action priors through a *bottom-up* perspective. For example, (Yan et al., 2024) view the LLM as a Bayesian prior over actions, enabling more focused exploration by restricting the candidate action space. (Zhang et al., 2024) similarly employ value-based filtering of LLM-proposed moves. (Liu et al., 2024b) use LLMs to both propose actions and model

world dynamics to carry out rollouts. These methods are orthogonal to ours and could be used in tandem. While these approaches reduce the search space, they typically do so state-by-state, treating the LLM as a single-step planner, rather than seeking multiple overarching strategies or explicitly balancing the value and specificity trade-offs that our top-down method targets. More critically, many such works are only applicable to tasks with text-format actions (e.g., 'open fridge', 'use key') that can be inapplicable for lower-level domains.

**Hierarchical RL** Several studies have explored hierarchical RL augmented by LLM-generated plans. SayCan (Ahn et al., 2022) and Video-Language Planning (Du et al., 2024) demonstrate that, if given a suitable set of action primitives or a text-to-video planning system, an LLM can sequence sub-goals. Language Planner (Huang et al., 2022) and Prog-Prompt (Singh et al., 2022) adopt similar frameworks but similarly rely on predefined macro-actions or prompts for decomposing tasks. These methods remain fundamentally *bottom-up* in that LLM suggestions must be hand-aligned to a fixed set of learned or engineered primitives, which usually requires either text-format action environments or extensive training of a language-conditioned model, which partially defeats the purpose in high-stakes scenarios. In contrast, our framework's strategies need not be limited to any single level of abstraction or a subset of primitives; and it flexibly explores multiple possible routes before committing to a refined direction.

**Reward shaping** Another line of work tackles reward engineering using LLMs. (Sarukkai et al., 2024), (Ma et al., 2024), and (Xie et al., 2024) show how language models can write code that shapes rewards. Although such approaches mitigate purely sparse rewards, they still rely heavily on trial-and-error adjustments of the reward logic by training models. In contrast, our method embeds reward shaping *within* a top-down strategizing process, automatically translating strategic knowledge into sub-goal rewards zero-shot *without* requiring trial and error in the environment or large-scale manual interventions.

While these prior works demonstrate that LLMs can offer valuable inductive biases—either by constraining action spaces, generating sub-goals, or shaping rewards—they generally omit the notion of *strategic exploration* that can fundamentally reduce sample complexity. By synthesizing top-down strategies as distributions over policy families and translating them into shaped rewards, our approach strategically narrows the search space without sacrificing expressivity, making a step toward real-world RL where extensive trial and error is not required.

We elaborate on the distinction with these and additional works across RL, Curriculum Learning, Hierarchical RL, LLM planning, as well as LLM reasoning in § D.

# 4. The Strategist Agent

We introduce the Strategist agent as our initial solution to the Strategy Problem. This agent follows the top-down approach to RL, operating in three key phases. First, it generates and explores potential strategies expressed in natural language, following a tree-based structure (§ 4.1 and 4.2). Second, it evaluates and prioritizes these strategies, iteratively refining them to achieve the necessary level of specificity (§ 4.3). Finally, it uses the best strategies to train RL agents through reward shaping, effectively guiding the policy search process toward the chosen strategies (§ 4.4). While this approach offers a promising direction for addressing the Strategy Problem, it represents just one of many possible solutions that merit future investigation.

## 4.1. Strategy Tree Construction

The Strategist's core component is the Strategy Tree, a hierarchical structure where each node represents a distribution over policies through natural language descriptions.

**Overview of the Strategist** Let $h_0$ denote the root node corresponding to the overall objective $G$. The tree expands through a refinement process in which each child $h_{i,j} \in \mathcal{C}(h_i)$ provides a more detailed specification of its parent's $h_i$ policy distribution. Formally, each node $h_i$ is associated with a distribution $\delta^{h_i} \in \Delta(\Pi_{\mathcal{M}})$ in the policy space $\Pi_{\mathcal{M}}$. Child nodes are constrained to represent increasingly specific policy subsets of their parents, with the branching factor reflecting the number of distinct approaches considered for decomposing the node's goal.

This recursive refinement process continues until the nodes reach sufficient *specificity* and *value* to serve as an effective inductive bias for the learning agent.

## 4.2. Node Types: Strategic Decomposition

Building on the structure of the Strategy Tree, the nodes $h_i$ can be categorized into two fundamental types that enable different forms of strategic reasoning:

1. **Approach Nodes:** These nodes branch into $\{h_{i,j}\}_{j=1}^m$ *mutually exclusive strategies* to achieve the parent's goal (e.g., "hunt" vs. "farm" to collect meat). Each child distribution $\delta^{h_{i,j}}$ narrows the scope of the policy by specifying implementation details. In practice, only the most promising approach branch is pursued.

2. **Plan Nodes:** These nodes decompose the parent goal into a sequence (or set) of *complementary sub-goals* that must all be completed (e.g., "craft a sword → locate animals → hunt"). The distribution $\delta^{h_i,\text{plan}}$ factors in these subgoals, and the overall properties of the node are derived from the composition of its components.

This duality between *vertical branching* (alternative approaches) and *horizontal decomposition* (sequential subgoals) enables the Strategy Tree to perform flexible, multi-level strategic reasoning.

## 4.3. Node Attributes: Value, feasibility and specificity

To select and prioritize exploration in the strategy tree according to the strategy problem (§ 2), each node $h_i$ is assigned two attributes—*value*, and *feasibility*, to check that the strategy is realizable in the environment. Note that *specificity*, the other key property of the Strategy Problem, is implicitly captured by the hierarchical structure of the tree. Hence, by design, the nodes satisfy $H(\delta^{h_{i,j}}) < H(\delta^{h_i})$.

**Value** $\mathcal{V}(h_i)$ captures how helpful the node's strategy is for achieving the parent objective. Formally, if $J(\pi)$ is the expected return of a policy $\pi$, then

$$\mathcal{V}(h_i) \approx \frac{\max_{\pi \in \text{supp}_\varepsilon(\delta^{h_{i,j}})} J(\pi)}{\max_{\pi \in \text{supp}_\varepsilon(\delta^{h_i})} J(\pi)}.$$

**Feasibility** $f(h_i)$ describes the likelihood that a policy from $\delta^{h_i}$ can actually be implemented by the RL agent in the environment. Formally, $f(h_i) = \delta^{h_i}(\Pi_{\text{agent}})$,, that is, the probability that a random draw from $\delta^{h_i}$ lies within the set of policies $\Pi_{\text{agent}}$ implementable by our learning system (e.g., limited action space, time constraints, etc.).

We detail aggregation, expansion, and selection in B.1.

## 4.4. Reward Shaping from Strategies

A key innovation of our framework is *strategy-guided reward shaping*, which translates textual strategies into a *quantitative* reward to guide RL agents. Concretely, given a strategy $h^\star$, corresponding to a root-to-leaf path of the Strategy Tree, we instantiate a reward $r_{h^\star}(s, a)$ that augments the original environmental reward. It proceeds in two steps:

1. **Node-Specific Scoring:** For the chosen strategy $h^\star$, we prompt an LLM with relevant state details $\{s_t\}$ and the textual specification of the node. The LLM outputs a scalar score $u_t$, indicating how "close" the state is to satisfying that sub-goal. We normalize $u_t$ to obtain the reward $\tilde{R}(s_t)$. For efficiency, we distill $\tilde{R}$ into a student network to reduce LLM calls.

2. **Aggregated Shaping Reward:** We combine the LLM-derived reward $\tilde{R}(s, a) = \mathbb{E}_{s' \sim P(s,a)}[\tilde{R}(s')]$ with the environment's original reward $R(s, a)$. $\alpha \in [0, 1]$ controls the strength of the strategy signal. In practice, $\alpha$ decays with training, progressively shifting to the reward of the environment:

$$r_{h^\star}(s_t, a_t) = (1 - \alpha)R(s_t, a_t) + \alpha \tilde{R}(s, a). \quad (1)$$

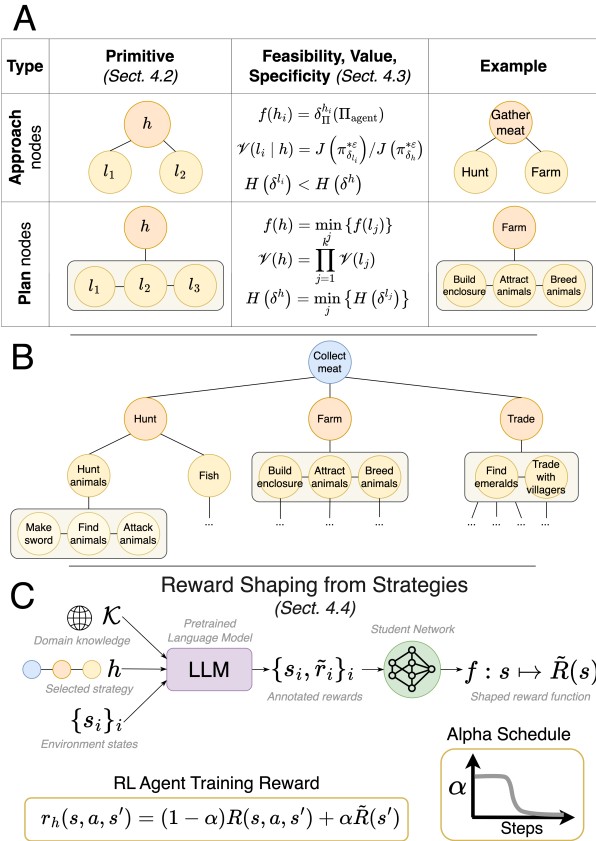

*Figure 2.* (A) Building blocks of the *Strategy Tree*: option nodes are different possible paths to reach the parent goal, while sequential nodes are sets of sub-goals that need to be sequentially completed to reach the parent goal. (B) Example of a possible Strategy Tree for the goal of maximizing meat collection in Minecraft. (C) Illustration of our *Strategy-based Reward Shaping* process.

In environments where actions are sufficiently high-level for the LLM, such as text-based environments, one can also consider using LLM-proposed actions as a policy regularizer (Yan et al., 2024; Zhang et al., 2024), and condition them on the strategy, thereby further accelerating convergence. However, we prefer to preserve applicability to low-level settings; therefore, we opt not to include this in this work. More details about this approach and the reward shaping process are provided in B.

## 5. Experiments

We evaluate the Strategist framework on different versions of Crafter (Hafner, 2021) and with different RL backbones: Proximal Policy Optimization (PPO) (Schulman et al., 2017), and Exploration via Distributional Ensemble (EDE) (Jiang et al., 2022). Our experiments demonstrate how top-down strategy generation and reward shaping significantly enhance sample efficiency and performance.

### 5.1. Environmental Setup

We utilize two main environment configurations:

**Modified Crafter (Easy & Medium)**   We modify the task to *meat collection*, and introduce additional mechanics that allow multiple plausible strategies. Specifically, cows can be bred via saplings, which can in turn be obtained from grass with probability $\frac{1}{2}$. We disable nighttime and hostile mobs, and we implement two difficulty tiers:

1. *Crafter-Easy*: cows require only one hit to kill and a single sapling to breed.
2. *Crafter-Medium*: cows require three hits to kill without a sword (one with a sword) and two saplings to breed.

In these modified versions, rewards are exclusively granted for collecting meat, thereby introducing sparseness: it may take many steps for an uninformed agent to discover how to kill or breed cows effectively. Thus, we view this as a rich test bed for examining strategic exploration.

**Original Crafter**   We also evaluate our framework in the original Crafter environment, which features 22 achievements, and the agent receives rewards for unlocking each new achievement. This setting presents a different sparse reward challenge and requires strategic prioritization across a wider range of potential goals (e.g., tool crafting, combat, resource gathering, survival needs).

### 5.2. Experimental Procedure

For each environment, we follow three stages:

1. **Strategy Tree Generation**. We run the Strategist agent using GPT-4o (OpenAI et al., 2024) to build a Strategy Tree (4.1) relevant to the overarching goal of the environment (e.g. maximize meat or achievements).

2. **Strategy Selection and Reward Shaping**. We extract promising strategies (the top two for Modified Crafter and four for Original Crafter, based on feasibility and value). For each strategy, we distill the LLM-based reward with a ResNet-18 (He et al., 2015) on 5,000 states. We combine the learned reward with the original environment one, gradually annealing $\alpha$ from $1.0$ (or $0.9$) to $0$ or $0.1$ during the course of training.

3. **RL Agent Training**. For each selected strategy, we train $N$ separate RL agents (typically $N = 8$ for PPO, $N = 3$ for EDE due to computational cost) for a set number of steps (e.g., $2 \times 10^6$ for Modified Crafter and $1 \times 10^6$ for Original Crafter). We use PPO (Schulman et al., 2017) as the primary RL backbone. For further validation, we also employ EDE (Jiang et al., 2022) and DreamerV3 (Hafner et al., 2023) as stronger baselines, demonstrating that the Strategist framework can be paired with different underlying RL algorithms

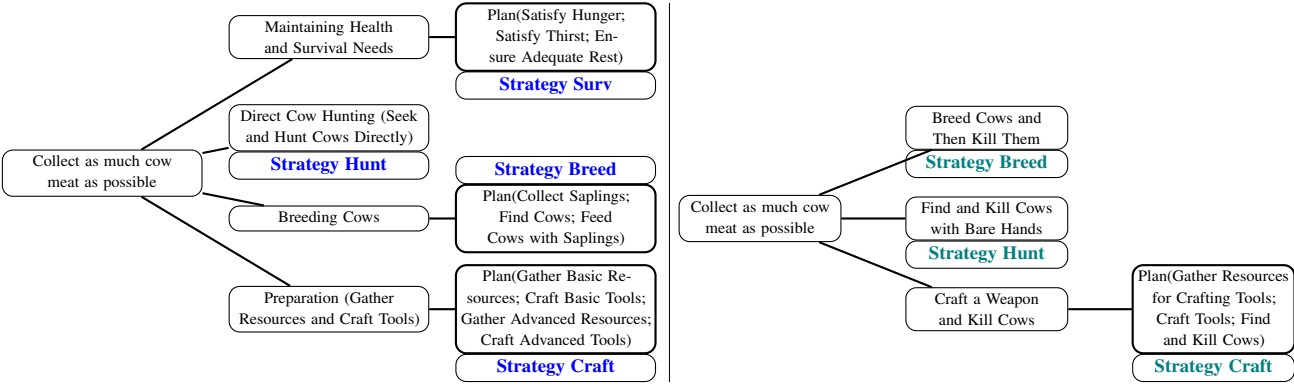

Figure 3. Strategy Tree obtained by running the Strategist agent (Section 4) with GPT-4o for the *Crafter-Easy* (*left*) and *Crafter-Medium* (*right*) environments. The strategies are ordered according to the product of feasibility and value estimated by the Strategist at each node.

(e.g., Strategist+PPO, Strategist+EDE). Baseline agents (PPO, EDE, DreamerV3) are trained under the same conditions but without shaped rewards ($\alpha = 0$).

The experimental setup is provided in detail in § B.

## 5.3. Results on Modified Crafter Environment

### 5.3.1. STRATEGY TREES

Figure 3 shows the results of the Strategist in the two modified environments, illustrating how it identifies multiple high-level approaches before refining them into concrete sub-goals. For instance, in *Crafter-Medium*, the Strategist proposes to hunt cows by crafting weapons first, as well as an alternative breeding-centered plan. Each node is annotated with an LLM-estimated *feasibility* (likelihood that the agent can achieve the strategy) and *value* (degree to which it might produce high returns).

### 5.3.2. PERFORMANCE WITH PPO

Figure 4 compares the average episode reward of the top two strategy-guided PPO agents (one focusing on hunting, the other on breeding) against a vanilla PPO baseline on *Crafter-Medium*. Both strategy-guided agents achieve faster and more meat collection following their strategies, which underscores the benefit of top-down guidance.

Table 1 further quantifies this by listing the number of steps required to achieve key milestones. "Hunting" is defined as "a > 0 average reward"; and "Breeding" as "> 1 average saplings given to a cow". Both strategy-guided PPO agents achieve hunting in 8/8 runs, typically within 600-900k steps, whereas PPO achieves this in only half of the runs and takes 1.35M steps on average. Breeding is achieved only by the breeding-guided agent, indicating that the strategist is effective in eliciting more complex behaviors.

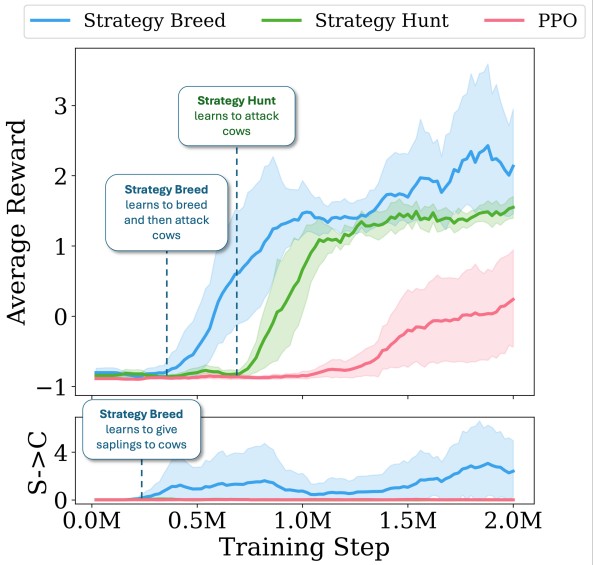

Figure 4. **Performance of strategy-guided RL agents (PPO)**. *Top panel*: Average episode reward on *Crafter-Medium*. *Bottom panel*: Average number of saplings given to cows per episode (S→C = "Saplings given to Cows"). Each line shows the average of N=8 runs, with error bars showing 95% confidence intervals.

### 5.3.3. EMPIRICAL STRATEGY METRICS WITH PPO

Next, we measure the feasibility and value in deployment when using PPO, shown for *Crafter-Easy* in Table 2. Feasibility is operationalized as the maximum normalized shaping reward the agent achieves, while value is measured by the maximum environment reward. We observe that for some strategies (e.g., breeding in *Crafter-Medium*), the realized feasibility and value exceed the initial estimates of the Strategist. For others (e.g., crafting-based plans), the observed performance is lower than anticipated. Figure 5 visually compares LLM-predicted metrics with post-training PPO measurements, revealing a strong overall correlation

| Model | Achieve Hunting Strategy | | Achieve Breeding Strategy | |
|---|---|---|---|---|
| | Successful runs | Step (M) | Successful runs | Step (M) |
| PPO | 4/8 | 1.35 (+/- 0.13) | 0/8 | - |
| Strategy Hunt | **8/8** | 0.89 (+/- 0.12) | 0/8 | - |
| Strategy Breed | **8/8** | **0.59 (+/- 0.17)** | 3/8 | **1.12 (+/- 0.76)** |

*Table 1.* Number of steps taken by PPO agents to reach Hunting and Breeding stages in the Crafter-Medium environment.

but some notable miscalibrations. These results suggest that while the Strategist can effectively propose strategies that boost performance, its heuristic estimates of feasibility and value can be refined through iterative feedback from the environment.

| Strategy | Estimated | | | | Observed | |
|---|---|---|---|---|---|---|
| | Feas. | Val. | Prod. | R. | Feasibility | Value |
| Strategy Hunt | **1.00** | **1.00** | **1.00** | **1** | 0.74 (+/- 0.03) | 17.18 (+/- 7.52) |
| Strategy Breed | 0.70 | 0.80 | 0.56 | 2 | **0.77 (+/- 0.02)** | **19.44 (+/- 2.68)** |
| Strategy Craft | 0.80 | 0.60 | 0.48 | 3 | 0.51 (+/- 0.01) | 3.86 (+/- 4.06) |
| Strategy Surv | 0.60 | 0.40 | 0.24 | 4 | 0.55 (+/- 0.01) | 2.02 (+/- 0.41) |

*Table 2.* Estimated vs. observed metrics for different strategies with PPO on *Crafter-Easy*. Feas.: Feasibility, Val.: Value, Prod.: Product, R.: Rank.

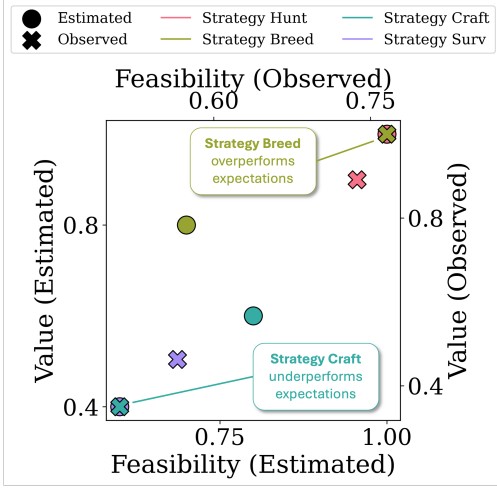

*Figure 5.* **Comparison of the LLM-estimated vs empirically-observed feasibility and values (PPO) for in the *Crafter-Easy* environment**. Observed values are normalized by the maximum observed value.

5.3.4. INTEGRATION WITH ADVANCED ALGORITHMS

As a stronger baseline RL algorithm, we run DreamerV3 (Hafner et al., 2023) with 50M parameters in the Modified Crafter environment. DreamerV3 performs slightly better than vanilla PPO on *Crafter-Medium*, but Strategist+PPO still outperforms it (see § C.5 for detailed DreamerV3 results). We also compare to two LLM-based baselines: LLM-as-a-policy, and Eureka (Ma et al., 2024) (see § C.8 in the appendix).

To further assess the Strategist's utility, we pair it with EDE, an advanced RL algorithm that achieves competitive results on the Crafter leaderboard[2]. As shown in Figure 6, Strategist+EDE, particularly with the "Breed" strategy, significantly outperforms vanilla EDE on *Crafter-Medium*. The increase in episode length for vanilla EDE suggests that its reward gains stem from improved survival. In contrast, the Strategist-guided agent consistently learns to breed cows -as indicated by the "sampling to cow" metric-, a complex behavior vanilla EDE struggles with, leading to a substantial increase in episode rewards. This indicates that even advanced exploration mechanisms benefit from a top-down strategic direction.

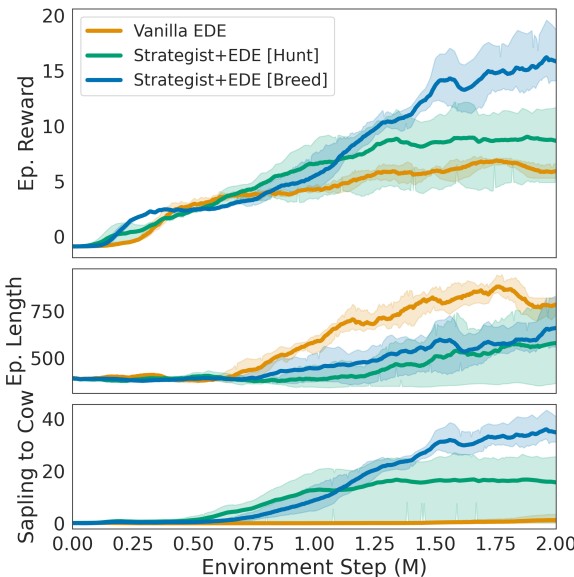

*Figure 6.* **Results of EDE and Strategist+EDE on Crafter-Medium.** *Top*: Episode reward. *Middle*: Episode length. *Bottom*: Average number of saplings given to cows per episode (Sapling to Cow). The Strategist+EDE outperforms vanilla EDE with both the hunting and breeding strategies. Error bars show the 95% confidence intervals across N=3 runs.

### 5.4. Results on Original Crafter Environment

In the original Crafter environment, the Strategist identifies four overarching strategies. Fight (combat and exploration), Craft (tool making), Resource (resource collection), and Needs (management of survival needs). We evaluate these by pairing them with the EDE agent.

Figure 7 shows that the Strategist+EDE variants focusing on Fight, Resource, and Craft significantly outperform vanilla EDE in terms of overall rewards (cumulative achievements). The Needs-guided strategy performs comparably with vanilla EDE, as basic survival achievements are easily reached by the baseline algorithm.

---

[2]https://github.com/danijar/crafter

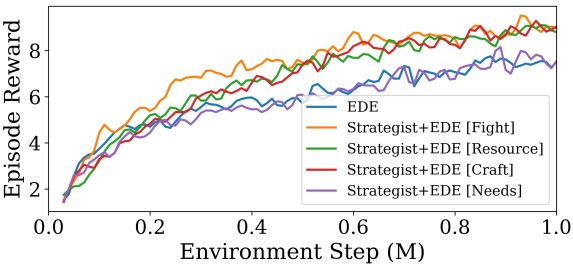

*Figure 7.* **Strategist+EDE on Original Crafter.** Episode Reward for vanilla EDE and Strategist+EDE with different strategies. Strategist+EDE with Fight, Resource, and Craft significantly outperform vanilla EDE.

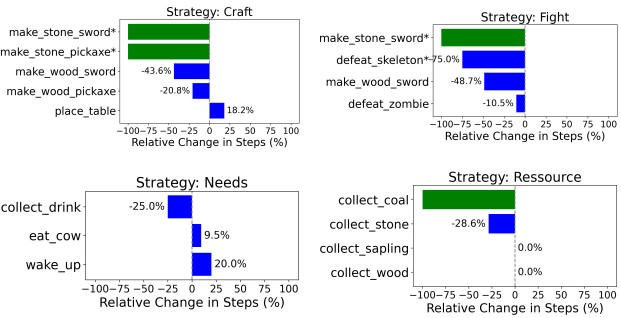

*Figure 8.* **Relative efficiency of Strategist+EDE variants across relevant achievements on the original Crafter environment.** Each subplot shows the change in number of environment steps required to complete an achievement related to each strategy $N$ times ($N=100$ by default, $N=10$ for rare achievements marked with *), relative to vanilla EDE. Green bars indicate the baseline failed to complete the achievement within 1M steps. Each strategy significantly improves efficiency for its related skills, demonstrating effective specialization.

### 5.4.1. QUALITATIVE BEHAVIOR AND SPECIALIZATION

The benefits of strategic guidance are more evident in the specialized behaviors of the agents. Figure 8 illustrates the relative efficiency gains in different achievements: Strategist+EDE agents require significantly fewer steps to complete achievements aligned with their given strategy compared to vanilla EDE. For instance, the Craft-guided agent is better at creating tools, and the Fight-guided agent excels at achieving combat-related objectives.

A detailed visualization of the signature of the achievements reached by each strategy compared to the baseline is shown in Figure 16. Each Strategist+EDE variant focuses and successfully unlocks achievements relevant to its high-level strategy. This qualitative evidence suggests that agents are indeed adopting behaviors consistent with the abstract strategies provided by the Strategist, leading to more effective exploration and task completion in this complex setting.

## 6. Key Takeaways and Outlook

**Why *Top–Down* Matters.** Our work reframes exploration in RL as a strategic design problem: rather than searching the entire policy space, we first search *over distributions of policies*. The resulting *Strategy Problem* formalism and its instantiation in the STRATEGIST agent deliver three conceptual advances:

1. **A principled objective.** By explicitly optimizing the *specificity–value* trade-off (Thm. 2.1), we provide a motivating sample–complexity result that scales with the *entropy of a strategy* rather than the state–action space.

2. **LLM-powered structure discovery.** LLMs are used *offline* to inject high-level domain knowledge, producing a Strategy Tree. This separates the quality of strategic guidance from the availability of low-level language-conditioned controllers, potentially expanding its applicability to robotics or healthcare workflows.

3. **End-to-end integration.** Our reward-shaping mechanism converts pure text into dense signals that any off-the-shelf RL algorithm can exploit. We demonstrate *plug-and-play* gains with PPO and EDE, showing consistent improvements in convergence speed, final return, and behavioral diversity.

**Empirical Significance.** Across three variants of Crafters, Strategist-guided agents (i) master sparse tasks *faster* than baselines, (ii) unlock qualitatively new behaviors (e.g., farming) that baselines never discover, and (iii) continue to add value when combined with state-of-the-art RL methods. These results position strategic guidance as an *orthogonal* axis of progress, complementary to algorithmic or model-architecture advances.

**Limitations.** First, LLM attribute proxies (§ 5.3.3) can be miscalibrated, leading to overexploration of unproductive branches. Second, once $\alpha \to 0$ and the rewards of the environment dominate, agents can revert to simpler strategies that are not aligned with the intended strategy. We provide a more detailed discussion of limitations in § E.

**Pathways Forward.** To address these limitations, several promising directions emerge. Although we currently commit to one final strategy during training, allowing the agent to pivot among multiple strategies mid-training is a natural extension, particularly in nonstationary settings. Closing the loop by feeding back agent progress into strategy updates may help address model miscalibration. Further opportunities include making the Strategist more scalable and integrating action-level guidance. A detailed discussion of these future work directions can be found in § E.

With top-down RL, we do not learn to plan by acting — we *act* because we have already planned.

## Acknowledgements

We extend our gratitude to the anonymous reviewers, area and program chairs, and members of the van der Schaar lab for their valuable feedback and suggestions. ML acknowledges sponsorship and support from AstraZeneca. AB acknowledges funding from Eedi. This work was supported by Azure sponsorship credits granted by Microsoft's AI for Good Research Lab. This work was supported by Microsoft's Accelerate Foundation Models Academic Research Initiative.

## Impact Statement

We propose a top-down RL framework that explicitly uses human-inspired strategies to guide exploration, bridging the gap between purely bottom-up methods and real-world decision making in complex domains like healthcare, logistics, and policy. By structuring high-level plans and sub-goals, our approach can reduce sample complexity, mitigate costly mistakes, and uncover beneficial behaviors more quickly.

However, translating domain knowledge into strategies through LLM brings new considerations. If an LLM's estimates of feasibility or value are miscalibrated—or if the underlying knowledge contains hidden biases—an agent may be steered toward suboptimal or even harmful plans. Robust oversight and validation of LLM outputs are therefore key, especially in sensitive applications.

Despite these risks, top-down RL has the potential to improve interpretability, safety, and adaptability in settings where traditional trial-and-error is impractical. By incorporating strategic knowledge from LLMs (or experts, if deemed necessary), the method can deliver more targeted exploration, potentially enabling faster convergence on high-value solutions in real-world environments that demand rigor and efficiency.

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

# A. Foundational Bounds for the Policy-Identification Meta-Algorithm

The results collected in this section justify why, under a suitably *specific* prior over policies and a simple probability threshold, one can narrow the search to a finite, provably small subset of candidate policies. The concrete algorithm we ultimately evaluate in the main paper is *different* and does not rely directly on the lemmas below; the present derivations serve only to illuminate the sample-complexity trade-offs that motivate our design choices.

## Entropy, probability thresholds, and support size

Let $\delta_\Pi$ be a probability distribution over a (finite or countably–infinite) collection of policies $\{\pi_i\}_{i \in \mathcal{I}}$. All logarithms are natural. Assume a bounded (Shannon) entropy

$$H(\delta_\Pi) \;=\; -\sum_{i \in \mathcal{I}} \delta_\Pi(\pi_i) \ln \delta_\Pi(\pi_i) \;\leq\; \Omega_M,$$

where $\Omega_M \geq 0$ is a user–specified *specificity budget*. For any threshold $\tau \in (0, 1]$ set

$$S_\tau \;=\; \big\{\pi_i : \delta_\Pi(\pi_i) \geq \tau\big\}.$$

**Lemma A.1** (Entropy–threshold support bound). *For every $\tau \in (0, 1]$,*

$$|S_\tau| \;\leq\; \min\Big\{\tfrac{1}{\tau}, \; 2 + \tfrac{\Omega_M}{\tau \ln(1/\tau)}\Big\}. \quad (*)$$

*The second term in $(*)$, $2 + \frac{\Omega_M}{\tau \ln(1/\tau)}$ (where the fraction is taken as $\infty$ if $\tau = 1$ and $\Omega_M > 0$, and as $0$ if $\Omega_M = 0$ making the term $2$), shows the impact of the entropy budget. Reducing $\Omega_M$ tightens this part of the bound. This second term is strictly smaller than $1/\tau$ if $\tau < 1/2$ and $\Omega_M < (1 - 2\tau) \ln(1/\tau)$.*

*Proof.* Since each $\pi \in S_\tau$ has mass at least $\tau$, $1 \geq \sum_{\pi \in S_\tau} \delta_\Pi(\pi) \geq \tau |S_\tau|$, so $|S_\tau| \leq 1/\tau$.

*Heavy and light atoms.* Partition $S_\tau$ into

$$H \;=\; \{\pi \in S_\tau : \delta_\Pi(\pi) > 1/e\}, \quad L \;=\; S_\tau \setminus H.$$

Because $\sum_{\pi \in H} \delta_\Pi(\pi) \leq 1$ and $\delta_\Pi(\pi) > 1/e$ on $H$, $|H| < e$ and hence $|H| \leq 2$.

*Entropy of light atoms (only if $L \neq \varnothing$).* If $L \neq \varnothing$, its elements $\pi$ satisfy $\tau \leq \delta_\Pi(\pi) \leq 1/e$. The condition $\delta_\Pi(\pi) \leq 1/e$ implies that $\tau \leq 1/e$ is necessary for $L$ to be non-empty. The function $f(x) = -x \ln x$ has derivative $f'(x) = -\ln x - 1$. For $x < 1/e$, $f'(x) > 0$, so $-x \ln x$ is increasing on $(0, 1/e)$. For every $\pi \in L$ we have $\tau \leq \delta_\Pi(\pi) \leq 1/e$. Therefore

$$-\delta_\Pi(\pi) \ln \delta_\Pi(\pi) \;\geq\; -\tau \ln \tau \;=\; \tau \ln(1/\tau).$$

If $\tau = 1/e$, $\tau \ln(1/\tau) = 1/e$. If $\tau < 1/e$, $\tau \ln(1/\tau)$ is positive. Hence, for $\tau \in (0, 1/e]$, if $L \neq \varnothing$:

$$\Omega_M \;\geq\; \sum_{\pi \in L} -\delta_\Pi(\pi) \ln \delta_\Pi(\pi) \;\geq\; |L| \, \tau \ln(1/\tau),$$

so $|L| \leq \Omega_M / [\tau \ln(1/\tau)]$. If $\tau > 1/e$, then $L = \emptyset$, so $|L| = 0$. In this case, the inequality $|L| \leq \Omega_M / [\tau \ln(1/\tau)]$ also holds (as $0$ is less than or equal to a non-negative quantity, assuming $\tau \ln(1/\tau) > 0$ for $\tau \in (0, 1)$).

*Combine.* If $L = \emptyset$ (e.g., if $\tau > 1/e$), then $|S_\tau| = |H| \leq 2$.

If $L \neq \emptyset$ (which requires $\tau \leq 1/e$), then

$$|S_\tau| = |H| + |L| \leq 2 + \Omega_M / [\tau \ln(1/\tau)].$$

Thus, the bound $|S_\tau| \leq 2 + \Omega_M / [\tau \ln(1/\tau)]$ holds for all $\tau \in (0, 1)$. Taking the minimum with the normalisation bound $1/\tau$ completes the proof for $\tau \in (0, 1)$. For $\tau = 1$: $\delta_\Pi(\pi_i) = 1$ means $\pi_i \in H$, so $L = \emptyset$. $|S_1| = |H|$. Since probabilities sum to 1, at most one policy can have probability 1, so $|S_1| \leq 1$. The bound formula $(*)$ yields $\min\{1, \infty\} = 1$ if $\Omega_M > 0$, and $\min\{1, 2\} = 1$ if $\Omega_M = 0$. $\square$

**Implications.** At most two policies can individually hold probability $> 1/e$, which explains the constant 2. Entropy alone cannot guarantee finite $|S_\tau|$ for an arbitrarily small $\tau$ if the set of policies is infinite; but in conjunction with a positive floor $\tau$, it yields a tractable candidate set whose size is explicitly controlled by $\Omega_M$ and $\tau$, as shown in the lemma.

### A.1. PAC Identification from a Finite Set of Policies

Consider a finite collection of candidate policies $\{\pi_1, \ldots, \pi_P\}$. Executing $\pi_i$ for one episode of length $K$ yields a return $R_i \in [0, 1]$; and since the episodes are independent and each episode starts from the same initial-state distribution, within each policy the returns are i.i.d. Let

$$\mu_i = \mathbb{E}[R_i], \qquad \mu^\star = \max_{1 \leq i \leq P} \mu_i.$$

**Uniform–sampling algorithm.** Fix accuracy $\varepsilon \in (0, 1]$ and confidence $\delta \in (0, 1)$. Sample each policy

$$m = \left\lceil \frac{2}{\varepsilon^2} \ln\left(\frac{4P}{\delta}\right) \right\rceil \quad \text{episodes,}$$

compute the empirical means $\hat{\mu}_i = \frac{1}{m} \sum_{t=1}^m R_i^{(t)}$, and return the policy $\hat{\pi} = \pi_{\arg\max_i \hat{\mu}_i}$.

**Proposition A.2** (Confidence PAC guarantee). *The uniform–sampling algorithm outputs a policy $\hat{\pi}$ such that*

$$\mu_{\hat{\pi}} \geq \mu^\star - \varepsilon$$

*with probability at least $1 - \delta$. The total number of episodes is $N = P\, m = \mathcal{O}\big(\frac{P}{\varepsilon^2} \ln(\frac{P}{\delta})\big)$.*

*Proof.* For any fixed $i$, Hoeffding's inequality gives

$$\Pr\left(|\hat{\mu}_i - \mu_i| > \tfrac{\varepsilon}{2}\right) \leq 2\exp\left(-\tfrac{1}{2}m\varepsilon^2\right) \leq \frac{\delta}{2P},$$

where the last step follows from the definition of $m$. A union bound over all $P$ policies ensures that, with probability at least $1 - \delta$, the event $|\hat{\mu}_i - \mu_i| \leq \varepsilon/2$ holds simultaneously for every $i$. On this event

$$\hat{\mu}_{\hat{\pi}} \geq \hat{\mu}_{i^\star} \geq \mu^\star - \tfrac{\varepsilon}{2}, \quad \mu_{\hat{\pi}} \geq \hat{\mu}_{\hat{\pi}} - \tfrac{\varepsilon}{2} \geq \mu^\star - \varepsilon.$$

The episode count is $N = P\, m$, completing the proof. $\square$

**Instance–dependent refinement.** One can significantly tighten the bound by making it gap-dependent, eliminating policies that are clearly suboptimal earlier and not sampling all policies uniformly (Even-Dar et al., 2006).

### A.2. From Episodic to Discounted Infinite Horizon

Many sequential–decision problems use a policy by its *discounted return*

$$G_\pi = \sum_{t=0}^\infty \gamma^t r_t, \qquad \gamma \in (0, 1), \ r_t \in [0, 1].$$

Because the sum is infinite, we truncate after $T$ steps and control the bias and variance this incurs.

**Lemma A.3** (Confidence estimation of discounted value). *Let $\{\pi_1, \ldots, \pi_P\}$ be a finite set of policies in a discounted MDP with factor $\gamma \in (0, 1)$. Fix accuracy $\varepsilon \in (0, 1]$ and confidence $\delta \in (0, 1)$.*

*Truncation horizon. Choose*

$$T \geq T^\star(\varepsilon, \gamma) = \left\lceil \frac{\ln\left(\frac{2}{\varepsilon(1-\gamma)}\right)}{\ln \frac{1}{\gamma}} \right\rceil \quad \implies \quad \frac{\gamma^T}{1 - \gamma} \leq \frac{\varepsilon}{2}.$$

*Sampling rule. For each policy collect $m$ independent trajectories of length $T$, where*

$$m \geq \frac{2}{\varepsilon^2(1-\gamma)^2} \ln\left(\frac{4P}{\delta}\right). \tag{$\star$}$$

***Guarantee.*** *Let $R_i^{(t)} = \sum_{\ell=0}^{T-1} \gamma^\ell r_\ell^{(t)}$ be the truncated return on trajectory $t$ for $\pi_i$, and set $\hat\mu_i = \frac{1}{m}\sum_{t=1}^m R_i^{(t)}$. Then, with probability at least $1 - \delta$,*

$$\left|\hat\mu_i - \mu_i\right| \le \varepsilon \quad \textit{for all } i = 1, \dots, P,$$

*where $\mu_i = \mathbb{E}[G_{\pi_i}]$ is the true infinite–horizon value.*

*Proof.* **Bias.** Since rewards lie in $[0, 1]$,

$$0 \le G_{\pi_i} - R_i^{(t)} = \sum_{k=T}^\infty \gamma^k r_k^{(t)} \le \frac{\gamma^T}{1-\gamma} \le \frac{\varepsilon}{2},$$

so $|\mathbb{E}[R_i^{(t)}] - \mu_i| \le \varepsilon/2$.

**Variance.** Each truncated return satisfies $0 \le R_i^{(t)} \le (1 - \gamma^T)/(1-\gamma) \le 1/(1-\gamma)$. Hoeffding's inequality with range $R = 1/(1-\gamma)$ gives, for any $i$,

$$\Pr\left(\left|\hat\mu_i - \mathbb{E}[R_i^{(t)}]\right| > \tfrac{\varepsilon}{2}\right) \le 2\exp\left(-2m\left(\tfrac{\varepsilon}{2}\right)^2(1-\gamma)^2\right) \le \frac{\delta}{2P},$$

where the last step uses $(\star)$. A union bound over the $P$ policies yields $|\hat\mu_i - \mathbb{E}[R_i^{(t)}]| \le \varepsilon/2$ for every $i$ with probability at least $1 - \delta/2$.

**Combine.** On the intersection of the bias and variance events we have $|\hat\mu_i - \mu_i| \le \varepsilon$ for all $i$. Because the variance event already occurs with probability $\ge 1 - \delta/2 \ge 1 - \delta$, the stated guarantee follows. $\square$

**Implication for identification.** Running SUCCESSIVE-ELIMINATION (Section A.1) on trajectories of length $T^\star$ and allocating the sample budget $(\star)$ in place of the episodic bound preserves all guarantees. The gap–independent complexity becomes

$$\tilde{O}\left((1-\gamma)^{-2}P\varepsilon^{-2}\right).$$

## A.3. High Specificity & Near-Optimal Value $\Rightarrow$ Small Sample Complexity

**Problem setting.** Consider an MDP $\mathcal{M}$ whose optimal value $J(\pi^\star)$ is attained by policy $\pi^\star$. For the episodic case, episode returns are normalised to $[0, 1]$, so $J(\pi^\star) \le 1$. For the discounted infinite-horizon case with $r_t \in [0, 1]$, $J(\pi^\star)$ may be up to $1/(1-\gamma)$. Auxiliary knowledge $\mathcal{K}$ provides a prior distribution $\delta_\Pi \in \Delta(\Pi_\mathcal{M})$. Given tolerances $(H_{\max}, v_{\min}, \tau)$ with $H_{\max} \ge 0$ and $v_{\min}, \tau \in (0, 1]$, assume

(a) **Specificity** : $H(\delta_\Pi) \le H_{\max}$;

(b) **Value** : there exists a policy $\pi^\dagger$ such that $\delta_\Pi(\pi^\dagger) \ge \tau$ and $J(\pi^\dagger) \ge J(\pi^\star) - v_{\min}$.

**Thresholded support.** With probability floor $\tau$ define the candidate set

$$S_\tau = \left\{\pi : \delta_\Pi(\pi) \ge \tau\right\}.$$

Lemma A.1 gives the deterministic bound

$$|S_\tau| \le P_\tau := \min\left\{\frac{1}{\tau},\ 2 + \frac{H_{\max}}{\tau\ln(1/\tau)}\right\}. \tag{2}$$

Since $\pi^\dagger \in S_\tau$, it holds that $J(\pi^\star_{S_\tau}) := \max_{\pi \in S_\tau} J(\pi) \ge J(\pi^\dagger) \ge J(\pi^\star) - v_{\min}$. Theorem A.4 demonstrates that by applying UNIFORM-SAMPLING to $S_\tau$ with an algorithm accuracy parameter $\varepsilon_{alg} = v_{\min}/2$, a policy $\hat\pi$ can be identified that is $(3/2)v_{\min}$-optimal relative to $J(\pi^\star)$ (i.e., $J(\hat\pi) \ge J(\pi^\star) - (3/2)v_{\min}$).

**Theorem A.4** (Specificity and value yield PAC efficiency). *Let $J(\pi^\star_{S_\tau}) = \max_{\pi \in S_\tau} J(\pi)$. Under assumption (b), $J(\pi^\star_{S_\tau}) \ge J(\pi^\star) - v_{\min}$. Run UNIFORM-SAMPLING (Proposition A.2) on the set $S_\tau$ with accuracy parameter $\varepsilon = \frac{v_{\min}}{2}$ and confidence $\delta \in (0, 1)$. The algorithm outputs $\hat\pi$ such that $J(\hat\pi) \ge J(\pi^\star_{S_\tau}) - \varepsilon$ with probability at least $1 - \delta$. Consequently, $J(\hat\pi) \ge J(\pi^\star) - (3/2)v_{\min}$ with probability at least $1 - \delta$.*

**Episodic returns.** *If each episode return lies in $[0,1]$ (hence $J(\pi)$ values are in $[0,1]$), the total number of episodes is*

$$N \;\leq\; C\,P_\tau\,\frac{\ln(P_\tau/\delta)}{\varepsilon^2}, \quad (\textit{where } \varepsilon = v_{\min}/2)$$

*for a universal constant $C \geq 8$.*

**Discounted infinite horizon.** *With discount factor $\gamma \in (0,1)$ and per-step rewards $r_t \in [0,1]$ (hence $J(\pi)$ values can be up to $1/(1-\gamma)$), let $\varepsilon_{lem} = \varepsilon/2 = v_{\min}/4$ be the accuracy target for Lemma A.3. Using trajectories of length $T^\star(\varepsilon_{lem}, \gamma)$ (from Lemma A.3, with its accuracy parameter set to $\varepsilon_{lem}$) and budget $m$ from Eq. $(\star)$ (also with its accuracy parameter set to $\varepsilon_{lem}$) yields*

$$N \;\leq\; C'\,P_\tau\,\frac{\ln(P_\tau/\delta)}{(1-\gamma)^2\varepsilon^2}, \quad (\textit{where } \varepsilon = v_{\min}/2 \textit{ is the overall algorithm accuracy})$$

*for another universal constant $C' \geq 8$.*

*Proof.* **(i) Size of the candidate set.** Equation (2) gives $|S_\tau| \leq P_\tau$.

**(ii) Achieved Optimality.** As established, under assumption (b), $\pi^\dagger \in S_\tau$ and $J(\pi^\star_{S_\tau}) = \max_{\pi \in S_\tau} J(\pi) \geq J(\pi^\dagger) \geq J(\pi^\star) - v_{\min}$. The UNIFORM-SAMPLING algorithm is applied with its accuracy parameter (denoted $\varepsilon_{alg}$ in Proposition A.2) set to $\varepsilon = v_{\min}/2$. By Proposition A.2, the output $\hat{\pi}$ satisfies $J(\hat{\pi}) \geq J^\star_{S_\tau} - \varepsilon$ (i.e., $J(\hat{\pi}) \geq J^\star_{S_\tau} - v_{\min}/2$) with probability at least $1 - \delta$. Therefore, $J(\hat{\pi}) \geq (J(\pi^\star) - v_{\min}) - v_{\min}/2 = J(\pi^\star) - (3/2)v_{\min}$. Thus, the policy $\hat{\pi}$ is $(3/2)v_{\min}$-optimal relative to $J(\pi^\star)$.

**(iii) Episodic complexity.** Applying Proposition A.2 to $S_\tau$ (of size at most $P_\tau$) with an overall algorithm accuracy parameter $\varepsilon = v_{\min}/2$ yields the stated bound. Proposition A.2 itself requires mean estimates to be $\varepsilon/2 = v_{\min}/4$ accurate. The constant $C$ accounts for the coefficient in Proposition A.2 (which is $2/((\cdot)/2)^2 = 8$ effectively) and any effects of the ceiling function.

**(iv) Discounted case.** For the UNIFORM-SAMPLING algorithm to achieve an overall accuracy of $\varepsilon = v_{\min}/2$, it requires estimates of policy values $J(\pi_i)$ that are accurate to $\varepsilon/2 = v_{\min}/4$. Thus, Lemma A.3 is invoked with its accuracy parameter set to $\varepsilon_{lem} = v_{\min}/4$.

The number of trajectories $m$ per policy from Lemma A.3 is then

$$m \geq \frac{2}{\varepsilon_{lem}^2 (1-\gamma)^2} \ln(4P_\tau/\delta) = \frac{32}{v_{\min}^2 (1-\gamma)^2} \ln(4P_\tau/\delta).$$

The total number of trajectories is $N = P_\tau m$ $\qquad \square$

**Interpretation.** Low entropy compresses the candidate set through the $H_{\max}/[\tau \ln(1/\tau)]$ term in Eq. (2). This entropy-dependent component of the bound is smaller than $1/\tau$ if $H_{\max} < (1 - 2\tau)\ln(1/\tau)$ (which implies $\tau < 1/2$); otherwise, the $1/\tau$ cap may be tighter or equally constraining. Consequently the episode complexity scales *linearly* in the entropy-controlled support size $P_\tau$, *polynomially* in $1/\varepsilon$ (where $\varepsilon = v_{\min}/2$ is the overall algorithm accuracy parameter), and only *logarithmically* in $1/\delta$.

## B. Further Implementation Details

This appendix provides additional technical details to ensure clarity and reproducibility.

### B.1. Aggregation for Plan Node Attributes

Recall from the main text that a *plan node* decomposes a higher-level goal into multiple sub-goals. Let $h_{i,\text{plan}}$ be a plan node with children $\{h_{i,1}, \ldots, h_{i,m}\}$, each corresponding to a sub-goal.

**Feasibility.** For plan nodes, the overall feasibility can be estimated as the minimum feasibility among the sub-goals, reflecting the fact that all sub-goals must be achievable:

$$f\big(h_{i,\text{plan}}\big) \;=\; \min_{j \in \{1, \ldots, m\}} f\big(h_{i,j}\big).$$

In practice, to stabilize LLM predictions or incorporate domain knowledge, one could use a weighted combination of sub-goal feasibilities. However, the minimum operator is the simplest and most intuitive choice.

**Value.** Because each sub-goal contributes to the overall objective, the final value can be viewed as an *aggregated* measure of sub-goal values. One straightforward strategy is to assume independence and multiply sub-goal values:

$$\mathscr{V}\big(h_{i,\text{plan}}\big) \;=\; \prod_{j=1}^{m} \mathscr{V}\big(h_{i,j}\big).$$

Alternatively, an average or maximum can be used if the sub-goals are substitutable or if certain sub-goals are more critical than others. In our experiments, we primarily adopt the multiplicative approach.

**Specificity.** Plan nodes combine multiple sub-goals that the policy must execute. If $H(\delta^{h_{i,j}})$ measures the entropy of each sub-goal's policy distribution, we approximate the overall entropy by the sum (or another suitable aggregator) of sub-goal entropies:

$$H\big(\delta^{h_{i,\text{plan}}}\big) \;\approx\; \sum_{j=1}^{m} H\big(\delta^{h_{i,j}}\big).$$

This ensures that each sub-goal reduces the space of valid policies, consistent with a stricter overall plan.

Nonetheless, these are non-central elements in our work and, in fact, the strategy trees we obtain in practice usually place plan nodes as leafs.

### B.2. Tree Expansion and Selection

As described in the main text, the Strategist builds and refines its Strategy Tree in an iterative manner. Let $\mathcal{L}$ be the current set of leaf nodes; each leaf is either an *approach node* or a *plan node* that has not yet been fully expanded. To decide which leaf to expand next, we pick

$$h_{\text{next}} \;=\; \arg\max_{h_{i_m} \in \mathcal{L}} \Big[ f\big(h_{i_m} \mid h_0\big) \times \mathscr{V}\big(h_{i_m} \mid h_0\big) \Big],$$

where $h_0$ is the root node, and

$$f\big(h_{i_m} \mid h_0\big) \;=\; \prod_{j=1}^{m} f\big(h_{i_j} \mid h_{i_{j-1}}\big), \quad \mathscr{V}\big(h_{i_m} \mid h_0\big) \;=\; \prod_{j=1}^{m} \mathscr{V}\big(h_{i_j} \mid h_{i_{j-1}}\big).$$

Here, $f\big(h_{i_m} \mid h_0\big)$ represents the root-relative feasibility (likelihood that this path of strategies is implementable), and $\mathscr{V}\big(h_{i_m} \mid h_0\big)$ represents the root-relative value (potential return of following that path).

**Strategy Decomposition Prompting.** Once $h_{\text{next}}$ is chosen, we prompt the LLM to refine it. Depending on its node type:

- **Approach Node Expansion.** The LLM is asked for a set of mutually exclusive "approaches" that refine the parent node's goal (e.g., different ways to gather resources). We then evaluate each approach in terms of feasibility, value, and specificity.

- **Plan Node Expansion.** The LLM is asked for a list of sub-goals or steps that collectively form a plan. We then compute aggregated feasibility and value as described in §B.1.

**Termination Criteria.** Expansion continues until any leaf node meets certain specificity thresholds that are implicitly evaluated by the LLM:

$$H\big(\delta^{h_{\text{leaf}}}\big) \ \leq \ H_{\max}.$$

If multiple leaves reach the threshold, we select the leaf with the highest $f(\cdot) \times \mathscr{V}(\cdot)$ product as the final strategy.

- **Additional stopping conditions.** We limit the maximum depth of the tree to avoid too many queries, but the limit is not hit in our runs.

### B.2.1. How Entropy is Handled and Specificity is Achieved

In our framework, **specificity** is a key property of a strategy distribution $\delta_\Pi$, ensuring that the strategy concentrates on a relatively small set of promising policies, thereby reducing the search space for the RL agent. We quantify specificity via the entropy $H(\delta_\Pi)$ of the policy distribution associated with a strategy node. A strategy is considered "sufficiently specific" if its entropy $H(\delta_\Pi)$ is below a certain threshold $H_{max}$, as defined in § 2.

Within the STRATEGIST agent's implementation, an explicit numerical calculation of entropy $H(\delta_\Pi)$ for each node using the LLM is not performed during the tree construction phase. Instead, **specificity is primarily achieved by design through the hierarchical nature of the Strategy Tree**. As described in § 4.3, child nodes are intended to represent increasingly specific policy subsets of their parents. This means that by construction, the policy distribution $\delta^{h_{i,j}}$ associated with a child node $h_{i,j}$ should have a lower entropy than that of its parent $h_i$, i.e., $H(\delta^{h_{i,j}}) < H(\delta^{h_i})$.

The LLM is prompted (see § B.3) to generate sub-goals that are more concrete and narrower in scope than their parent goal. The threshold $H_{max}$ serves as a conceptual guide in the Strategy Problem formulation (§ 2). While not directly calculated and checked against $H_{max}$ by the LLM for each node, the process of decomposing goals into more detailed sub-goals implicitly drives the entropy down.

### B.2.2. How Value is Estimated

In the context of the Strategy Tree (§ 4.3), the **value** $\mathscr{V}(h_i)$ of a node $h_i$ is an estimate of how much of the parent node's potential optimal return is preserved by focusing on the strategy $\delta^{h_i}$ associated with node $h_i$. It's defined as the ratio of the maximum expected return achievable within the support of $\delta^{h_i}$ to the maximum expected return achievable within the support of its parent's strategy distribution $\delta^{h_{\text{parent}}}$. The term $v_{min}$ in the Strategy Problem formulation (§ 2) relates to this by ensuring that $J(\pi^{*\varepsilon}_{\delta_\Pi}) \geq (1 - v_{min})J(\pi^\star)$, meaning the best policy within the strategy's support is at least $(1 - v_{min})$ times as good as the true optimal policy $\pi^\star$.

During the construction of the Strategy Tree by the STRATEGIST agent, the **value attribute for each node is initially estimated by the LLM**. The LLM is prompted (see § B.3) to assess this based on its domain knowledge and the textual description of the goal associated with the node. This is a heuristic estimate. The STRATEGIST agent does not perform rollouts or direct calculation of $J(\pi)$ during the tree generation phase to determine these values. Instead, it relies on the LLM's zero-shot estimation capabilities. As discussed in § 5.3.3, these initial LLM estimates can be refined with empirical data.

### B.2.3. How Feasibility is Estimated

The **feasibility** $f(h_i)$ of a strategy node $h_i$ in the Strategy Tree quantifies the likelihood that the set of policies $\delta^{h_i}$ associated with that node can actually be implemented by the learning agent within the given environment and its constraints. As formally defined in § 4.3, $f(h_i) = \delta^{h_i}(\Pi_{\text{agent}})$, which is the probability mass of $\delta^{h_i}$ that falls within the set of policies $\Pi_{\text{agent}}$ implementable by the RL agent. This probability represents how much of the current strategy aligns with policies achievable by the agent's learning hypothesis class.

Similar to $\mathscr{V}$, the **feasibility attribute for each node is estimated by the LLM** (see § B.3) based on its understanding of the task and the environment.

B.2.4. HOW INITIAL LLM ESTIMATES CAN BE UPDATED

The initial estimates of feasibility and value for each node in the Strategy Tree are provided by the LLM in a zero-shot manner. These are heuristic estimates and can be imperfect.

As suggested in § 5.3.3, these initial LLM estimates can and ideally should be **dynamically updated based on feedback from the agent's actual interactions with the environment**. As the RL agent attempts to execute strategies and sub-goals, its performance (e.g., achieved rewards, completion rates) can be used to refine the 'feasibility' and 'value' attributes stored in the tree. This "closing the loop" by feeding back agent progress into strategy updates is highlighted as an important area for future work. The experimental setup presented primarily uses the initial LLM estimates for strategy selection but then empirically measures the realized feasibility and value to compare against these initial heuristics.

B.2.5. BUILDING THE STRATEGY TREE

We instantiate the Strategist agent using a GPT-4o model (OpenAI et al., 2024), using the prompts provided in § B.3. After building the entire Tree following the process detailed above and in § 4, we run another instance of GPT-4o to refine the estimates of feasibility and value using the prompt in Listing 2 below.

## B.3. LLM Prompts

**LLM Prompting Details.** Below are the different prompts used to query LLMs in our experiments.

```
1  strategy\_context: |
2    You are going to provide the high-level strategy for a goal by proposing different
         options and breaking the problem down into progressively lower-level subgoals.
3    The structure of goals and sub-goals will be organised in a tree structure, where
         the root node is the overall goal, and children of a node are different
         strategies, ie options of subgoals that can each help fulfill the parent node.
4    It is important to note that the children of a node represent different **
         independent** strategies. Each branch should be a viable way of getting to the
         overall without needing any other branch of the tree to be completed.
5
6    Each node has two key measures:
7    - **Feasibility**: Confidence that this goal can be achieved, assessed
         independently of the parent.
8      - A feasibility of 1 means the goal is certain to be achieved.
9      - A feasibility of 0.5 reflects 50\% confidence in the goal's achievability.
10   - **Value**: How restrictive the present goal is compared to its parent in terms of
          their potential to maximise expected returns. Especially between possible sub-
         goals, the higher value subgoal should correspond to the one that is expected
         to maximise expected rewards for the overall objectve.
11     - A value of 1 means achieving this goal is as good as it gets for the parent
           goal.
12     - A value of 0.5 suggests that achieving this goal is expected to yield half the
           parent's potential returns.
13
14   The tree will be built iteratively, starting with the root node. At each step, the
         most promising leaf node will be selected based on the highest product of
         feasibility and value, and you'll be provided with its root-to-leaf path as a
         list of nodes.
15
16   You can one of the following but not both in the same response:
17   1. Break down the leaf node into children (use this only if you think the node
         should be broken down further, avoid unnecessary complexity), or
18   2. Leave it as a leaf node if it's specific enough to represent clear, low-level
         actions that can be easily executed.
19
20   If the root-to-leaf path includes sequential nodes, you will receive multiple leaf
         nodes. Apply the same process to each leaf, one after the other.
21
22   The root node (id=1) will already be in the tree. You will add children nodes to
         break it down. Before making decisions, reflect on:
23   1. Whether to break down the node,
```

```
24    2. Which nodes to add, and
25    3. How you assess their feasibility and value.
26
27    Use the appropriate command only when you are confident in your choice of subgoals
          and feasibility/value estimates.
28    Finish your response after giving your commands.
29    Do not rehearse your commands. Once you give them it should be the end of your
          response.
30
31
32  tree\_commands: |
33    ### Command Structure for Modifying the Tree
34
35    To execute a command, write it as a plain line without quotes or symbols. The
          available commands are:
36
37    - **ADD|[goal description]|[parent id (int)]|[feasibility]|[value]**: Add a new
          node.
38    - **MODIFY|[id (int)]|[new feasibility]|[new value]**: Modify the feasibility and
          value of an existing node.
39    - **REMOVE|[id (int)]**: Delete a node and all its children.
40    - **ADD\_SEQUENTIAL|"Goal 1;Goal 2;...;Goal N"|[parent id]|[feasibility1]|[value1
          ]|[feasibility2]|[value2]|...[feasibilityN]|[valueN]**: Add multiple sequential
           subgoals to the specified parent node. Goals are separated by semicolons. Make
           sure there are as many feasibility and values as there are goals.
41    - **READ\_TREE**: read the current tree
42    - **END\_TREE**: Use this if the current strategy cannot be broken down any further
          . Use this command only if you are **absolutely** sure that you want to end the
           generation process.
43    Note: do not use the character "|" inside the goal description.
44    Once you have executed a command or a set of command, finish your response. Do not
          write anything else after using a series of any of the commands above. You will
           have the opportunity to read the tree and make further modifications at the
          next iteration.
45
46  provide\_best\_node: |
47    Here is the current most promising root-to-leaf path in the tree:
48
49    {root\_to\_leaf\_path}
50
51    Please evaluate this path and choose to either:
52    1. Break down the leaf node into children (use this only if you think the node
          should be broken down further, avoid unnecessary complexity), or
53    2. Leave it as a leaf node if it represents clear, low-level actions that an agent
          can easily execute.
54
55    If you break down the node, explain your reasoning and use the ADD command to add
          new children.
56    If you leave it as a leaf node, justify why it's specific enough. Then, you can ask
           to read the tree with READ\_TREE, freely modify the nodes of your choice.
57    If you do not want to do any further changes to the tree, terminate the process
          with END\_TREE.
58
59    Ensure you reflect carefully about the factors that can influence the feasibility
          and value of any new nodes you add, and provide a final estimation.
```

*Listing 1.* Base Prompt for building the Strategy Tree

```
1  You will be provided with a tree showing strategic options towards an overall goal
       located in the root node. You will be asked to evaluate the feasibility and value
        of each node, based on the following definitions:
2  Feasibility: How likely is the current node's strategy can be successfully
       implemented in the environment? A score between 0 (very unlikely) and 1 (very
       likely).
```

```
3  Value: If the current node's strategy was achieved, how good would it be towards
      achieving **its parent goal**, relative to other potential strategies? A score
      between 0 (very bad strategy, e.g., even if achieved, achieving it does not imply
       fulfilling the parent goal) and 1 (optimal way to reach the parent goal and
      maximise the reward).A high-value node must be sufficient on its own to achieve
      the parent goal.The maximum value goal should correspond to the strategy that can
       yield the highest value for the parent goal.
4  **Make sure to evaluate each node strictly on the basis of the information provided
      as part of the node description**.
5  Do not evaluate nodes downstream of a node that has flag 'sequential: True'. Evaluate
       the sequential node as usual but ignore its downstream nodes.
6  For the root node (id=1), just set feasibility to 1 and value to 1.
7  For each node, reason step by step to reach your estimated feasibility and value.
8  Please format the final outcome between <answer> and </answer> as a dict: {'id': {'
      feasibility':.., 'value':..}, ..}
```

*Listing 2.* Value and Feasibility Metrics

```
1  Given the goal '{goal}', analyze each of the {n\_states} states provided by
      describing how close or far it is from achieving the goal.
2
3  Based on this analysis, assign a score between 0 and 5 to each state, where:
4     - 0: The state is very far from the goal.
5     - 1: The state is far from the goal.
6     - 2: The state is somewhat far from the goal.
7     - 3: The state is somewhat close to the goal.
8     - 4: The state is close to the goal.
9     - 5: The state is very close to the goal.
10
11 **Examples of State Characteristics:**
12    - **Score 0 or 1:**
13       - No cows in sight.
14       - No saplings for breeding.
15       - Low health due to unmet survival needs.
16
17    - **Score 2:**
18       - Survival needs are stable.
19       - Maybe saplings in inventory but no cows in sight.
20
21    - **Score 3 or 4:**
22       - One or more cows in sight.
23       - Health and survival metrics not critical.
24
25    - **Score 5:**
26       - A cow adjacent to the agent and saplings in inventory for breeding.
27
28 Only take into account characteristics that are directly related to the goal. For
      example, if the strategy involves breeding, take saplings into account in your
      scoring; if the goal is simply hunting, ignore saplings in your evaluation; only
      consider health and vitals if they are explicitly stated in the goal.
29 Do NOT repeat parts of the game description such as the different objects in the game
       or the game objective.
30 Only reflect on things that are directly related to the states and the goal.
31
32 Once you reach a conclusion, always give your final answer with the following format:
33 SCORES:<score for state 1>,<score for state 2>,...,<score for state {n\_states}>
34
35 For example, if scores of 1, 5 and 4 are assigned to state 1, 2 and 3 respectively,
      your answer would finish with: "SCORES:1,5,4".
36 Make sure to finish your response with this final line: "SCORES:"
```

*Listing 3.* Reward Shaper

```
1  goal\_context: |
2    The environment is: Crafter
3    The overall goal is: "Collect as much cow meat as possible. Each item of meat
         collected by killing a cow counts for 1 point. Your goal is to maximise the
         number of points collected within one episode of the environment."
4
5
6  game\_context: |
7    Crafter is an open-world survival game used as a reinforcement learning (RL)
         benchmark. The game focuses on evaluating an agent's general capabilities by
         presenting a procedurally generated, 2D environment where agents interact
         through visual inputs. Crafter is designed for deep exploration, long-term
         reasoning, and generalization, making it a rich environment for RL research.
8
9    # Environment Overview:
10   - World Generation: Each episode takes place in a procedurally generated 64x64 grid
          world. Terrain types include:
11     - Grasslands
12     - Forests
13     - Lakes
14     - Mountains
15     - Caves
16   The terrain determines the distribution of creatures and resources.
17
18   # Game Mechanics:
19   1. Survival Needs:
20     - Health: The player/agent has health points (HP), which are depleted by hunger,
             thirst, fatigue, or attacks from monsters. HP regenerate when hunger,
           thirst, and fatigue are satisfied.
21     - Hunger: Decreases over time; satisfied by eating meat (from cows) or plants.
22     - Thirst: Decreases over time; satisfied by drinking from lakes.
23     - Rest: Decreases over time; replenished by sleeping in safe areas.
24     - If any of these values (hunger, thirst, rest) reach zero, the player starts
           losing health.
25
26   2. Resources:
27     - Resources can be collected and used to craft tools. They include:
28       - Wood (from trees)
29       - Stone
30       - Coal
31       - Iron
32       - Diamonds
33       - Saplings (plants that can be grown for food)
34
35   3. Crafting and Technology Tree:
36     - Players can use a crafting table and a furnace to craft tools:
37       - Wood Pickaxe and Wood Sword
38       - Stone Pickaxe and Stone Sword
39       - Iron Pickaxe and Iron Sword
40     - Higher-tier tools like the iron pickaxe and sword require more advanced
           resources (iron, coal) and improve efficiency in collecting resources and
           defeating enemies.
41
42   4. Creatures:
43     - Cows: Source of meat (food). Can be found in grass areas.
44
45   # Game Interface:
46   - Observation: Agents receive 64x64 pixel images representing a top-down view of
          the world, including an 8x8 grid area surrounding the player and an inventory
          display (health, hunger, thirst, rest, collected materials).
47   - Action Space: A flat categorical space with 17 actions, including movement,
          interacting with objects (e.g., gathering resources or attacking), crafting
          tools, and placing objects.
48
49   # Other important comments:
```

```
50   Blocks that can be freely moved through are: grass, path, sand.
51   Blocks that can be placed and easily destroyed are: stone, tree, table, furnace.
52   Blocks that can be destroyed are: coal, iron, diamond, water.
53   Blocks that can neither be moved through nor destroyed are: lava.
54   Meat can only be collected by killing a cow. Cows can be found in grassy areas.
55   A cow is killed by hitting it 1 time. No weapons or tools are required to kill a
        cow.
56   Cows can be bred by feeding them 1 sapling. In such case, a new cow will spawn next
        to the first one.
57   The player can move in four directions: up, down, left, right.
```

*Listing 4.* Crafter Easy Environment

```
1   goal\_context: |
2     The environment is: Crafter
3     The overall goal is: "Collect as much cow meat as possible. Each item of meat
          collected by killing a cow counts for 1 point. Your goal is to maximise the
          number of points collected within one episode of the environment."
4
5   game\_context: |
6     Crafter is an open-world survival game used as a reinforcement learning (RL)
          benchmark. The game focuses on evaluating an agent's general capabilities by
          presenting a procedurally generated, 2D environment where agents interact
          through visual inputs. Crafter is designed for deep exploration, long-term
          reasoning, and generalization, making it a rich environment for RL research.
7
8     # Environment Overview:
9     - World Generation: Each episode takes place in a procedurally generated 64x64 grid
           world. Terrain types include:
10      - Grasslands
11      - Forests
12      - Lakes
13      - Mountains
14      - Caves
15     The terrain determines the distribution of creatures and resources.
16
17    # Game Mechanics:
18    1. Survival Needs:
19      - Health: The player/agent has health points (HP), which are depleted by hunger,
            thirst, fatigue, or attacks from monsters. HP regenerate when hunger,
            thirst, and fatigue are satisfied.
20      - Hunger: Decreases over time; satisfied by eating meat (from cows) or plants.
21      - Thirst: Decreases over time; satisfied by drinking from lakes.
22      - Rest: Decreases over time; replenished by sleeping in safe areas.
23      - If any of these values (hunger, thirst, rest) reach zero, the player starts
            losing health.
24
25    2. Resources:
26      - Resources can be collected and used to craft tools. They include:
27        - Wood (from trees)
28        - Stone
29        - Coal
30        - Iron
31        - Diamonds
32        - Saplings (plants that can be grown for food)
33
34    3. Crafting and Technology Tree:
35      - Players can use a crafting table and a furnace to craft tools:
36        - Wood Pickaxe and Wood Sword
37        - Stone Pickaxe and Stone Sword
38        - Iron Pickaxe and Iron Sword
39      - Higher-tier tools like the iron pickaxe and sword require more advanced
            resources (iron, coal) and improve efficiency in collecting resources and
            defeating enemies.
```

```
40
41     4. Creatures:
42        - Cows: Source of meat (food). Can be found in grass areas.
43
44     # Game Interface:
45     - Observation: Agents receive 64x64 pixel images representing a top-down view of
            the world, including an 8x8 grid area surrounding the player and an inventory
            display (health, hunger, thirst, rest, collected materials).
46     - Action Space: A flat categorical space with 17 actions, including movement,
            interacting with objects (e.g., gathering resources or attacking), crafting
            tools, and placing objects.
47
48     # Other important comments:
49     Blocks that can be freely moved through are: grass, path, sand.
50     Blocks that can be placed and easily destroyed are: stone, tree, table, furnace.
51     Blocks that can be destroyed are: coal, iron, diamond, water.
52     Blocks that can neither be moved through nor destroyed are: lava.
53     Meat can only be collected by killing a cow. Cows can be found in grassy areas.
54     By default, a cow is killed by hitting it 3 times. When using a sword however, only
            1 hit is required.
55     Cows can be bred by feeding them 2 saplings. In such case, a new cow will spawn
            next to the first one.
56     The player can move in four directions: up, down, left, right.
```

*Listing 5.* Crafter Medium Environment

### B.4. Reward Shaping Details

**Reward Shaping Network.** We use a ResNet-18 (He et al., 2015) to predict sub-goal progress scores for each environment state. Key architectural choices:

- **Input shape:** $64 \times 64$ image frames.

- **Output dimension:** continuous score $\tilde{R}(s) \in [-1, 1]$, using $\tanh$ as the final activation function.

- **Data Augmentation:** We augment the labeled dataset by a factor of 10 using mixup (Zhang et al., 2018).

- **Loss function:** Mean-Squared Error (MSE) loss.

- **Training setting:** We train the model for 10 epochs on the augmented dataset.

We backpropagate using the Adam optimizer (Kingma & Ba, 2017) with learning rate $1 \cdot 10^{-4}$, $\beta_1 = 0.9$, $\beta_2 = 0.999$.

**Data collection pipelines:** We collect 5000 frames: 4000 from 200 trajectories of PPO agents pretrained on the environment, and 1000 from the expert human demonstration dataset provided with the original Crafter environment (Hafner, 2021). Observations are converted to textual data using Smartplay (Wu et al., 2024). We iteratively sample 5 states, provide their textual descriptions to a GPT-4o-mini model, and use the prompt for reward shaping (provided above) to generate reward signals.

**Combining Rewards.** The final reward at time $t$ is

$$r_{h^\star}(s_t, a_t) \;=\; (1 - \alpha)\, R(s_t, a_t) \;+\; \alpha\, \frac{\tilde{R}(s_t)}{\beta},$$

with $\beta$ a normalization constant that we set to 10 to balance the shaped reward with the original reward $R$. We use an inverse square-root decay schedule for $\alpha$, either from 1 to 0 between 50k and 1M ("short schedule") or from 0.9 to 0 between 500k and 1.5M ("long schedule"). A piecewise or adaptive schedule could be employed if the environment exhibits dynamic difficulty or if sub-goals are naturally learned in stages.

## B.5. RL Algorithm Details

**PPO Hyperparameters.** In our experiments, we use the standard PPO (Schulman et al., 2017) implementation from `stable_baselines_3` (Raffin et al., 2021) with the default hyperparameters:

- **Policy/Value Network Architecture:** "CnnPolicy"

- **Rollout Length per Update:** 2048

- **Learning Rate:** $3 \cdot 10^{-4}$.

- **Batch Size:** 64

- **Clip Range:** 0.2

- **Discount Factor:** 0.99

**Training Schedules.** We train for a total of 2M steps, logging performance metrics and 20k and saving checkpoints every 500k steps). We evaluate each performance metrics over 10 episodes.

## B.6. Extended Environment Details

**Modified Crafter Setup.** We create a modified version of the Crafter environment (Hafner, 2021), with the following modifications:

- **Reward Function:** The reward corresponds to the number of meat items collected. We also include the original rewards of $-0.1$ when the player loses health. We discard all other rewards related to achievements in the original environment.

- **Breeding Mechanism:** Collecting a sapling and giving it to a cow spawns an additional cow nearby.

- **Increased :** Collecting a sapling and giving it to a cow spawns an additional cow nearby.

- **Sapling Collection Probability:** We increased the probability of collecting saplings from grass from 10% to 50%.

- **Discard night mechanism:** We discard the night mechanism, zombies and skeletons.

- **Difficulty Levels:** We implemented two difficulty levels:
    - *Crafter-Easy:* Cows die with one hit (whether or not a sword is used) and breed with one sapling.
    - *Crafter-Medium:* Cows die with three hits without a sword and one hit with a sword, and breed with two saplings.

## B.7. Code and hardware

Experiments were ran on two NVIDIA RTX 6000 ADA GPUs with 48GB VRAM and 120GB RAM.

To ensure reproducibility, we will provide the full source code upon acceptance.

# C. Extended Experimental Results

This appendix provides a more detailed look at our experimental results, complementing the analysis in § 5. We present learning curves across different environment difficulties and hyperparameter settings, provide quantitative metrics on how quickly agents achieve key strategic milestones (hunting and breeding), and compare the Strategist's initial LLM-based estimates of strategy feasibility and value against their empirically observed counterparts. These extended results underscore the robustness of our top-down approach, demonstrating its ability to accelerate learning and guide exploration effectively across various conditions, while also offering insights into the current capabilities and limitations of LLM-based strategy assessment.

## C.1. Environment Reward Over Optimization Steps

Figures 9 and 10 illustrate the learning dynamics of our strategy-guided agents compared to the PPO baseline across both *Crafter-Easy* and *Crafter-Medium* environments, using both short and long $\alpha$ schedules (controlling the transition from shaped to environment rewards).

In the *Crafter-Easy* setting (Figure 9), both strategy-guided agents generally learn faster than PPO, although PPO eventually catches up in terms of average reward. Notably, the "Strategy Breed" agent consistently achieves a much higher rate of giving saplings to cows (S→C), demonstrating its successful adherence to the breeding strategy, especially under the long alpha schedule (Figure 9b). The "Strategy Hunt" agent also shows some breeding behavior (Figure 9a), suggesting some overlap or discovery.

The benefits of strategic guidance become even more pronounced in the *Crafter-Medium* environment (Figure 10). Here, PPO struggles significantly, often failing to achieve substantial rewards. In contrast, both "Strategy Breed" and "Strategy Hunt" achieve significantly higher rewards and converge much faster. Crucially, only the "Strategy Breed" agent learns to breed cows effectively, highlighting how top-down guidance is essential for discovering these more complex, long-horizon behaviors in challenging, sparse-reward settings. The $\alpha$ schedule influences the learning speed, with the short schedule (Figure 10a) often leading to faster initial gains, particularly for "Strategy Breed".

Additionally, Figures 9 and 10 show the same results with 68% confidence intervals instead of 95%. The conclusions from each plot remain unchanged.

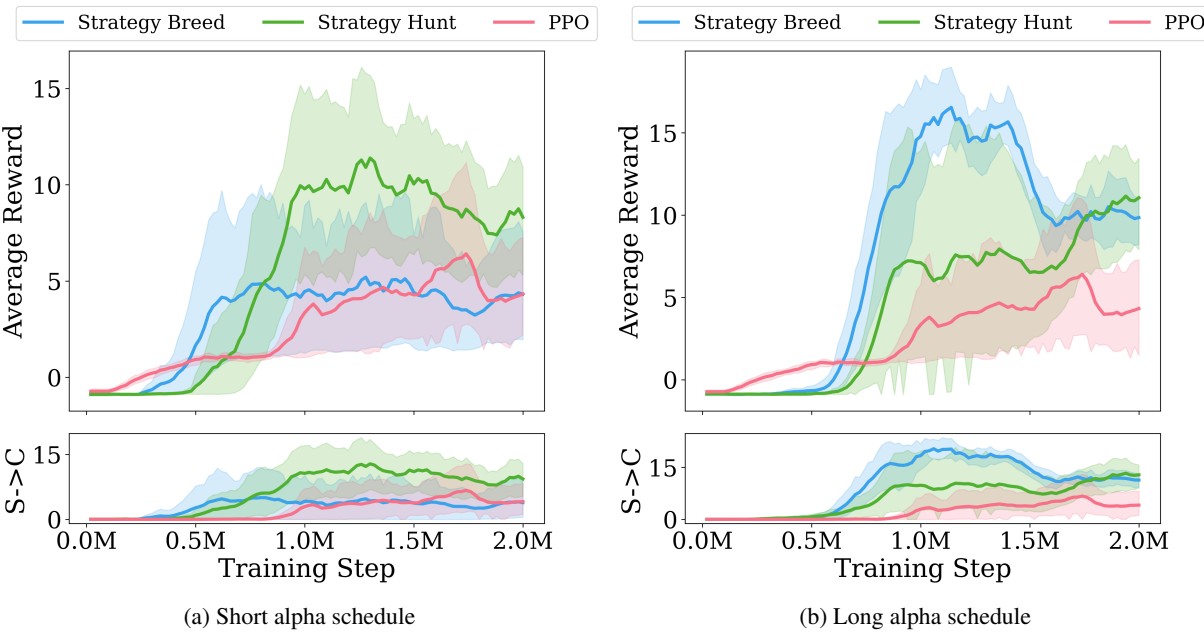

(a) Short alpha schedule

(b) Long alpha schedule

*Figure 9.* **Crafter-Easy Environment Results.** Average episode reward (top row of each subfigure) and saplings-to-cows (S→C, bottom row) for short (a) and long (b) alpha schedules. Each line shows the average of N=8 runs, with error bars showing 95% confidence intervals.

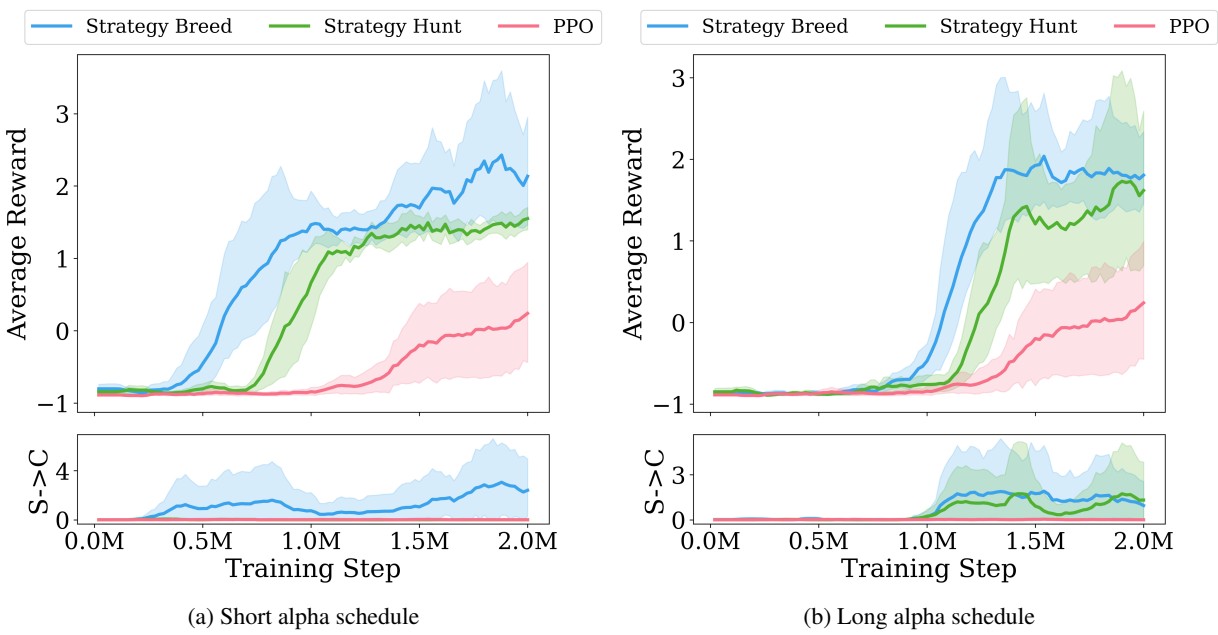

(a) Short alpha schedule

(b) Long alpha schedule

*Figure 10.* **Crafter-Medium Environment Results.** Average episode reward (top row of each subfigure) and saplings-to-cows (S→C, bottom row) for short (a) and long (b) alpha schedules. Each line shows the average of N=8 runs, with error bars showing 95% confidence intervals.

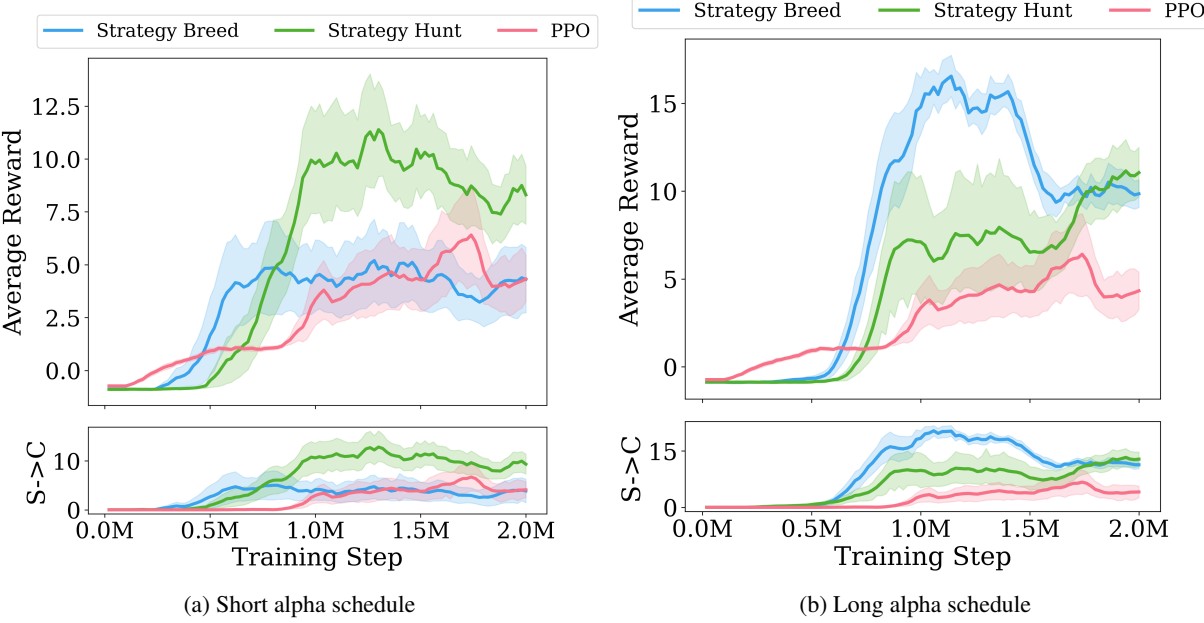

(a) Short alpha schedule

(b) Long alpha schedule

*Figure 11.* **Crafter-Easy Environment Results.** Average episode reward (top row of each subfigure) and saplings-to-cows (S→C, bottom row) for short (a) and long (b) alpha schedules. Each line shows the average of N=8 runs, with error bars showing 68% confidence intervals.

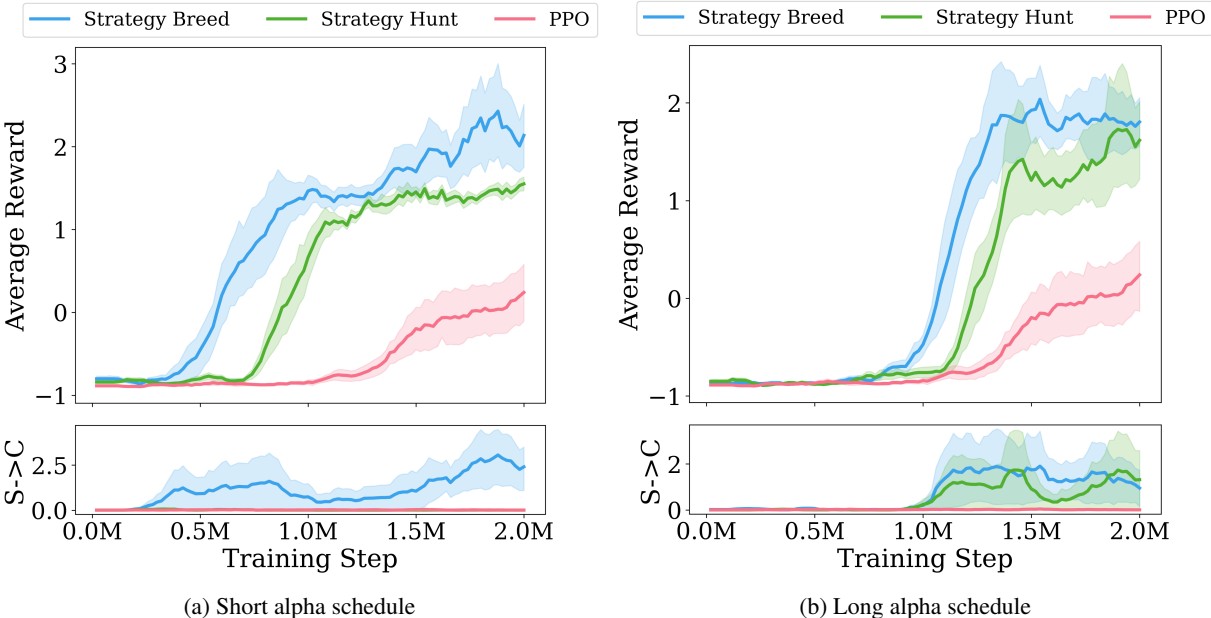

(a) Short alpha schedule

(b) Long alpha schedule

*Figure 12.* **Crafter-Medium Environment Results.** Average episode reward (top row of each subfigure) and saplings-to-cows (S→C, bottom row) for short (a) and long (b) alpha schedules. Each line shows the average of N=8 runs, with error bars showing 68% confidence intervals.

## C.2. Steps to Hunting and Breeding

To quantify the efficiency gains, Tables 3 through 6 report the number of training steps required for agents to consistently achieve "Hunting" (average reward $> 0$) and "Breeding" (average S→C $> 1$), along with the success rate across 8 runs.

These tables confirm the visual trends. In *Crafter-Medium* (Tables 5 and 6), PPO only succeeds in hunting in half its runs and never learns to breed. In stark contrast, both strategy-guided agents achieve hunting in almost all runs, often significantly faster than PPO. Most importantly, the "Strategy Breed" agent is the only one to achieve any consistent breeding success in the medium environment, even though its success rate remains below 100%, indicating the difficulty of this long-horizon task. The "Strategy Hunt" agent, when guided by the short alpha schedule, focuses solely on hunting and does not discover breeding, further emphasizing the specificity induced by our reward shaping.

| | Achieve Hunting Strategy | | Achieve Breeding Strategy | |
| --- | --- | --- | --- | --- |
| Model | Successful runs | Step (M) | Successful runs | Step (M) |
| PPO | 8/8 | 0.20 (+/- 0.08) | 3/8 | 1.00 (+/- 0.26) |
| Strategist—Breed...lings | 8/8 | 0.45 (+/- 0.10) | 3/8 | 0.87 (+/- 0.88) |
| Strategist—Direc...ctly) | 8/8 | 0.67 (+/- 0.12) | 6/8 | 0.68 (+/- 0.23) |

*Table 3.* Number of steps taken to reach Hunting and Breeding stages in the Crafter-Easy environment (short alpha).

| | Achieve Hunting Strategy | | Achieve Breeding Strategy | |
| --- | --- | --- | --- | --- |
| Model | Successful runs | Step (M) | Successful runs | Step (M) |
| PPO | 8/8 | 0.20 (+/- 0.08) | 3/8 | 1.00 (+/- 0.26) |
| Strategist—Breed...lings | 8/8 | 0.65 (+/- 0.15) | 8/8 | 0.65 (+/- 0.20) |
| Strategist—Direc...ctly) | 8/8 | 1.07 (+/- 0.41) | 7/8 | 1.03 (+/- 0.61) |

*Table 4.* Number of steps taken to reach Hunting and Breeding stages in the Crafter-Easy environment (long alpha).

| | Achieve Hunting Strategy | | Achieve Breeding Strategy | |
| --- | --- | --- | --- | --- |
| Model | Successful runs | Step (M) | Successful runs | Step (M) |
| PPO | 4/8 | 1.35 (+/- 0.13) | 0/8 | - |
| Strategist—Breed... Them | 8/8 | 0.59 (+/- 0.17) | 3/8 | 1.12 (+/- 0.76) |
| Strategist—Find ...Hands | 8/8 | 0.89 (+/- 0.12) | 0/8 | - |

*Table 5.* Number of steps taken to reach Hunting and Breeding stages in the Crafter-Medium environment (short alpha).

| | Achieve Hunting Strategy | | Achieve Breeding Strategy | |
| --- | --- | --- | --- | --- |
| Model | Successful runs | Step (M) | Successful runs | Step (M) |
| PPO | 4/8 | 1.35 (+/- 0.13) | 0/8 | - |
| Strategist—Breed... Them | 8/8 | 1.02 (+/- 0.09) | 3/8 | 1.29 (+/- 0.46) |
| Strategist—Find ...Hands | 7/8 | 1.15 (+/- 0.17) | 1/8 | 1.00 (+/- 0.00) |

*Table 6.* Number of steps taken to reach Hunting and Breeding stages in the Crafter-Medium environment (long alpha).

## C.3. Strategy Metrics: Estimated vs. Observed

We now examine how well the LLM's initial estimates of strategy feasibility and value align with the actual performance observed during training. Tables 7 through 10 present these comparisons. Feasibility and value are defined in § 5.

A consistent finding is that the LLM's initial ranking does not always perfectly predict the best-performing strategy. In *Crafter-Medium* (Tables 9 and 10), "Strategy Breed" was ranked third by the LLM but achieved the highest observed value and feasibility in both alpha settings. This suggests that while LLMs provide valuable heuristics, they can underestimate the potential of more complex strategies or overestimate the ease of others (like "Strategy Craft," which was ranked first but performed less well). This highlights the importance of potentially closing the loop, as discussed in § 6, allowing environmental feedback to refine these initial estimates.

| | Estimated | | | | Observed | |
|---|---|---|---|---|---|---|
| Strategy | Feasibility | Value | Product | Rank | Feasibility | Value |
| Strategy Hunt | **1.00** | **1.00** | **1.00** | **1** | 0.69 (+/- 0.02) | **13.82 (+/- 7.82)** |
| Strategy Breed | 0.70 | 0.80 | 0.56 | 2 | **0.70 (+/- 0.03)** | 6.48 (+/- 7.35) |
| Strategy Craft | 0.80 | 0.60 | 0.48 | 3 | 0.45 (+/- 0.01) | 6.19 (+/- 5.43) |
| Strategy Surv | 0.60 | 0.40 | 0.24 | 4 | 0.53 (+/- 0.00) | 2.29 (+/- 0.77) |

*Table 7.* Strategist metrics for *Crafter-Easy* environment, using a short alpha schedule.

| | Estimated | | | | Observed | |
|---|---|---|---|---|---|---|
| Strategy | Feas. | Val. | Prod. | R. | Feasibility | Value |
| Strategy Hunt | **1.00** | **1.00** | **1.00** | **1** | 0.74 (+/- 0.03) | 17.18 (+/- 7.52) |
| Strategy Breed | 0.70 | 0.80 | 0.56 | 2 | **0.77 (+/- 0.02)** | **19.44 (+/- 2.68)** |
| Strategy Craft | 0.80 | 0.60 | 0.48 | 3 | 0.51 (+/- 0.01) | 3.86 (+/- 4.06) |
| Strategy Surv | 0.60 | 0.40 | 0.24 | 4 | 0.55 (+/- 0.01) | 2.02 (+/- 0.41) |

*Table 8.* Strategist metrics for *Crafter-Easy* environment, using a long alpha schedule.

| | Estimated | | | | Observed | |
|---|---|---|---|---|---|---|
| Strategy | Feasibility | Value | Product | Rank | Feasibility | Value |
| Strategy Craft | 0.75 | **0.85** | **0.64** | **1** | 0.60 (+/- 0.02) | 1.83 (+/- 0.10) |
| Strategy Hunt | **0.90** | 0.50 | 0.45 | 2 | 0.63 (+/- 0.03) | 1.80 (+/- 0.15) |
| Strategy Breed | 0.60 | 0.70 | 0.42 | 3 | **0.68 (+/- 0.04)** | **2.95 (+/- 1.73)** |

*Table 9.* Strategist metrics for *Crafter-Medium* environment, using a short alpha schedule.

| | Estimated | | | | Observed | |
|---|---|---|---|---|---|---|
| Strategy | Feasibility | Value | Product | Rank | Feasibility | Value |
| Strategy Craft | 0.75 | **0.85** | **0.64** | **1** | 0.63 (+/- 0.02) | 2.38 (+/- 1.72) |
| Strategy Hunt | **0.90** | 0.50 | 0.45 | 2 | 0.66 (+/- 0.01) | 2.01 (+/- 1.85) |
| Strategy Breed | 0.60 | 0.70 | 0.42 | 3 | **0.72 (+/- 0.01)** | **2.49 (+/- 1.45)** |

*Table 10.* Strategist metrics for *Crafter-Medium* environment, using a long alpha schedule.

### C.4. Visualizing Estimated vs. Observed Strategy Metrics

Figures 13 and 14 provide a visual comparison between the LLM-estimated and empirically observed feasibility and value metrics for each strategy across different settings. Each point represents a strategy: circles denote the LLM's initial estimates, while 'X' marks show the values observed after training the RL agents. These plots allow us to assess the LLM's zero-shot prediction capabilities.

These scatter plots visually confirm the findings from the strategy metrics tables. While there is often a positive correlation—strategies estimated as highly feasible and valuable tend to perform well—there are notable and informative discrepancies. For example, in the *Crafter-Medium* environment with the long alpha schedule (Figure 14b), "Strategy Breed" (green) moves significantly up and to the right from its estimated position (circle) to its observed position (X), indicating it performed much better than predicted. Conversely, "Strategy Hunt" (pink) moves down and left, showing it was somewhat overestimated in that specific run.

These visualizations emphasize both the promise and the current limitations of using LLMs for zero-shot strategy evaluation. They provide valuable heuristics that can guide initial strategy selection, but they also highlight the need for incorporating environmental feedback to refine these estimates, motivating future work on iterative refinement and adaptive strategy selection.

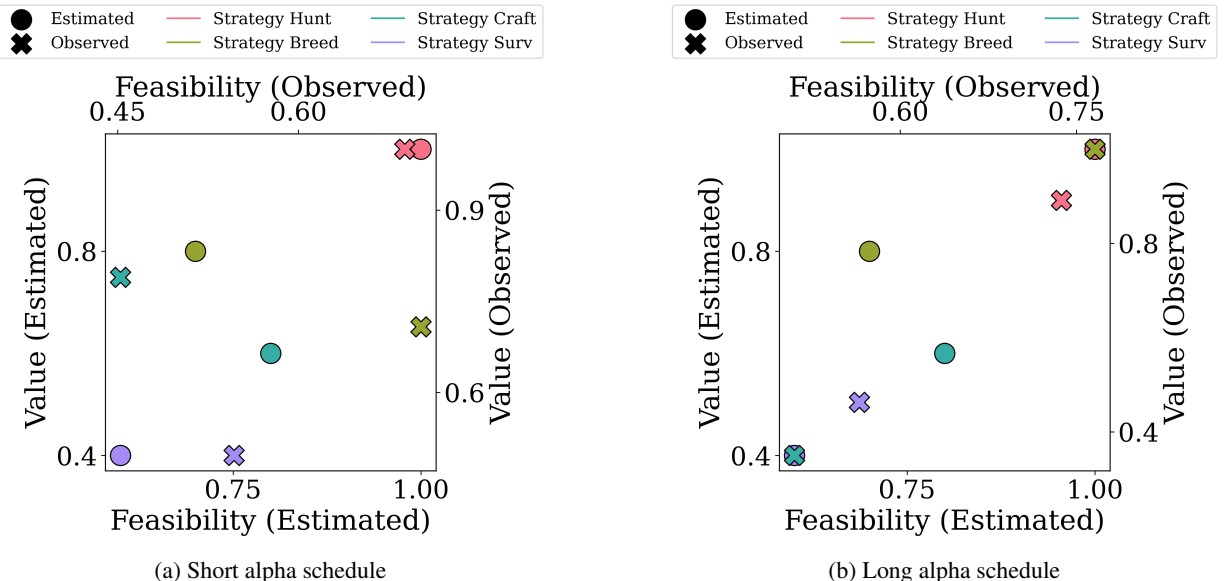

(a) Short alpha schedule           (b) Long alpha schedule

*Figure 13.* **Crafter-Easy: Estimated vs. Observed Metrics.** Comparison of LLM-estimated (circles) vs. empirically-observed (X marks) feasibility and value for the (a) short and (b) long alpha schedules. Observed values are normalized by the maximum observed value.

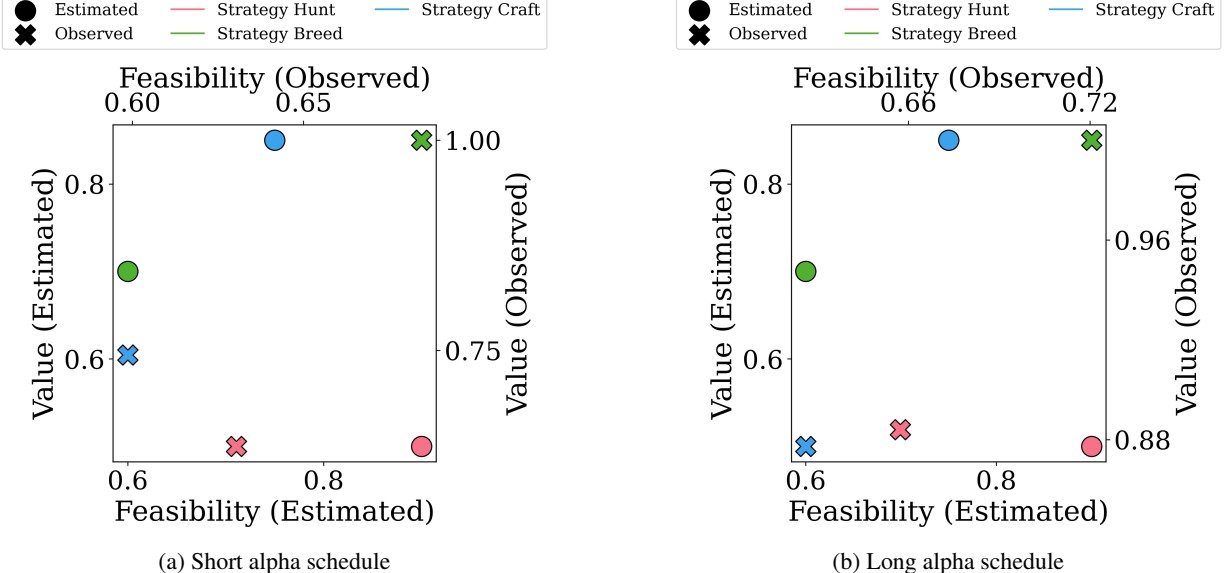

(a) Short alpha schedule

(b) Long alpha schedule

*Figure 14.* **Crafter-Medium: Estimated vs. Observed Metrics.** Comparison of LLM-estimated (circles) vs. empirically-observed (X marks) feasibility and value for the (a) short and (b) long alpha schedules. Observed values are normalized by the maximum observed value.

## C.5. Comparison with Stronger Baselines (DreamerV3) on Crafter-Medium

Figure 15 shows the performance of DreamerV3 (medium size, 50M parameters) on Crafter-Medium. The DreamerV3 baseline performs slightly better than vanilla PPO on *Crafter-Medium*, but Strategist+PPO still outperforms it.

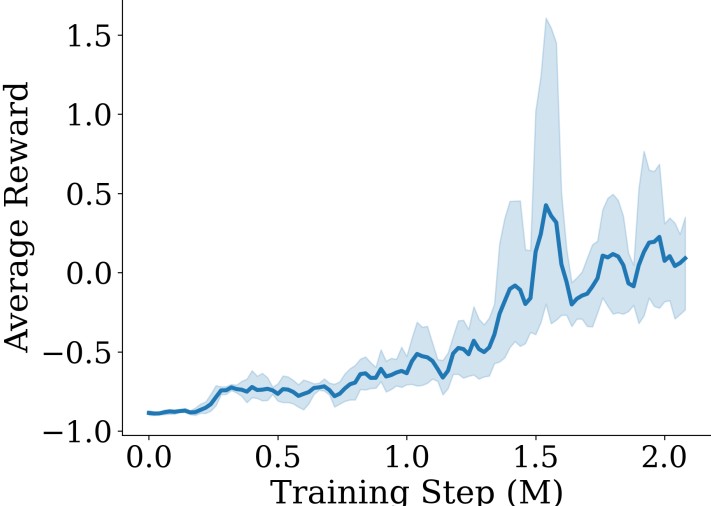

*Figure 15.* **DreamerV3 on Crafter-Medium.** We train a DreamerV3 (50M parameters) model on Crafter-Medium over 2M training steps. Results averaged over N=4 runs. Error bars show the 95% confidence intervals.

## C.6. Combining the Strategist with Stronger RL Agents (EDE) on Crafter-Medium

To demonstrate the Strategist's complementarity with strong, exploration-focused RL algorithms, we combined our framework with Exploration via Distributional Ensemble (EDE) (Jiang et al., 2023). We evaluated this combination on our *Crafter-Medium* task. Figure 6 in the main paper shows these results.

The results highlight that while vanilla EDE learns an effective survival-focused hunting strategy (indicated by increasing episode length), it fails to discover the more complex breeding strategy. In contrast, when guided by the Strategist, EDE+Breed not only discovers breeding (as shown by the Sapling to Cow metric) but achieves approximately double the final reward compared to vanilla EDE. This demonstrates that our top-down strategic guidance can unlock high-value, complex behaviors that even strong, modern exploration methods miss on their own.

## C.7. Performance on the Original Crafter Environment

To further assess the generalizability of our approach, we also evaluated Strategist+EDE on the original Crafter environment (Hafner, 2021), which features a diverse set of achievement-based rewards rather than a single 'collect meat' goal. The Strategist identified four distinct high-level approaches for maximizing achievements:

- **Fight (Combat and Exploration):** Manage combat with creatures and explore the environment to unlock achievements related to enemies and rare resources.

- **Craft (Tool Crafting):** Use collected resources to create essential tools needed for higher-tier activities.

- **Resource (Resource Gathering):** Focus on collecting primary resources required for various achievements.

- **Needs (Survival Needs Management):** Maintain hunger, thirst, rest, and health to ensure the agent can complete other goals efficiently.

Figure 7 shows the overall performance. The Fight, Resource, and Craft strategies significantly outperform vanilla EDE in terms of total achievements collected. The Needs strategy performs similarly to the baseline, likely because survival achievements are relatively easy to obtain through basic exploration, which EDE already does well.

To understand *how* these strategies achieve better performance, we analyzed their impact on specific achievements. Figure 8 illustrates the relative efficiency gains. For many achievements, especially rarer ones (marked with *), the guided strategies achieve them much faster than vanilla EDE, and in some cases (green bars), EDE fails to achieve them at all within the 1M step budget.

Finally, Figure 16 provides a compelling visual summary of this specialization. The heatmap clearly shows distinct "footprints" for each strategy. The Craft strategy excels at tool-making; Fight boosts combat and related crafting; Resource gathers materials like stone, coal, and iron (which vanilla EDE barely reaches); and Needs focuses on basic survival. This demonstrates the Strategist's ability to generate diverse and effective high-level plans that successfully steer strong RL agents toward specific, valuable behaviors, unlocking performance gains and enabling targeted skill acquisition in complex, multi-faceted environments.

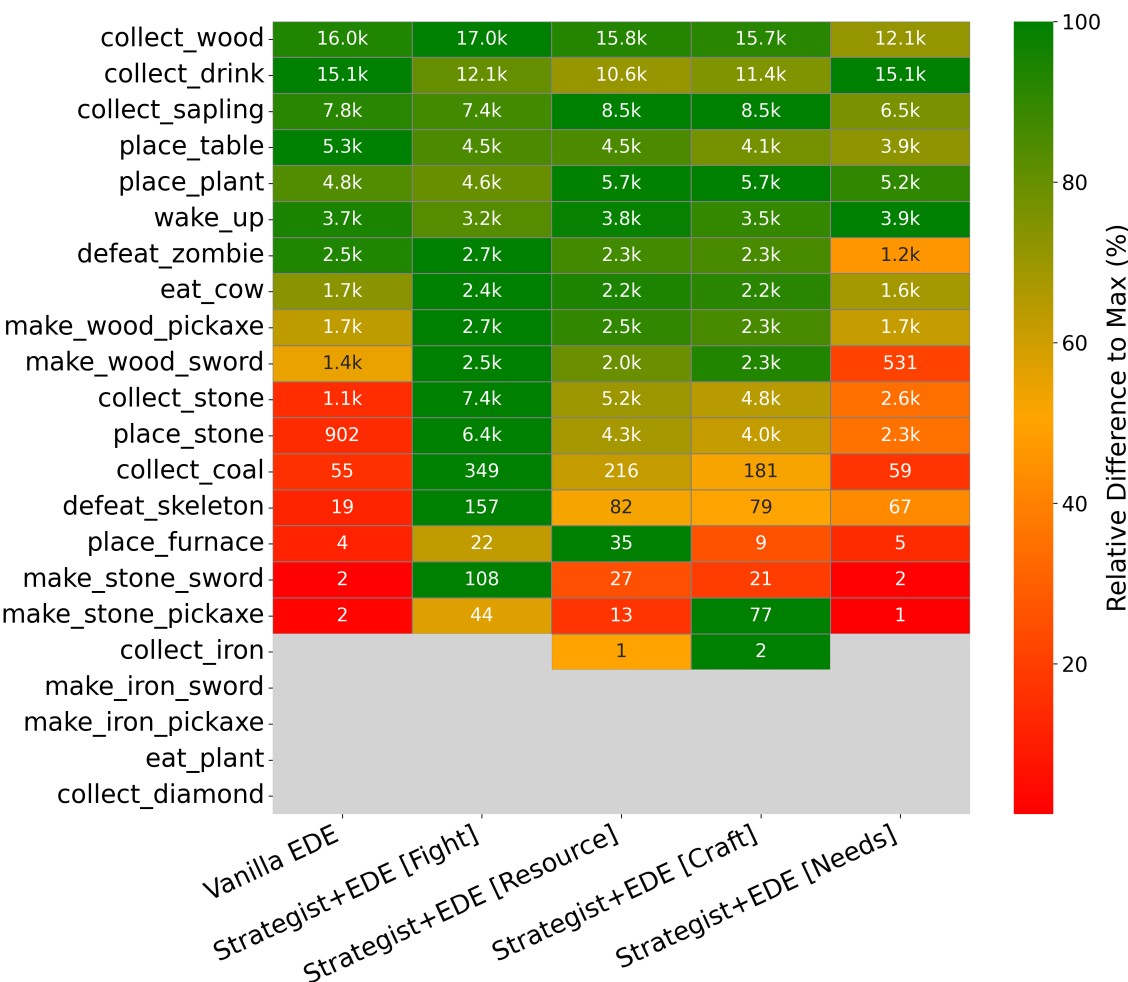

*Figure 16.* **Achievement completion counts for Vanilla EDE and different Strategist+EDE strategies on the original Crafter environment.** Each cell shows the number of times an achievement was completed within 1M training steps, colored by relative difference to the maximum per row. Gray indicates zero completions. Strategist+EDE variants show clear specialization: the Craft strategy excels at tool-related achievements (e.g.: make stone pickaxe and sword), Fight boosts combat outcomes (e.g.: make stone sword, defeat skeletons), Resource collects rare resources (e.g.: collect stone, coal, iron), and Needs supports survival needs (e.g.: drink, sleep).

## C.8. Comparison with LLM-Based Baselines

To further contextualize our contributions, we compared the Strategist framework against prominent LLM-based approaches that operate differently, either by using the LLM directly as a policy or by using it for iterative reward design. These comparisons were conducted on our *Crafter-Medium* environment.

### C.8.1. LLM-AS-POLICY.

We implemented a baseline where a GPT-4o-mini model was prompted at each timestep to select the next action directly. This represents a direct, a-strategic use of the LLM's world knowledge and reasoning for low-level control. This approach performed poorly, achieving an average reward of only $0.43 \pm 1.01$ (over $N = 15$ runs), similar to the baseline PPO at 2 million steps. Furthermore, the computational cost associated with LLM inference at every single step makes this approach impractical for many RL tasks.

### C.8.2. LLM-BASED REWARD DESIGN (EUREKA)

To contextualize the Strategist's approach, we conducted experiments with Eureka (Ma et al., 2023), an LLM-based reward design framework, on our modified Crafter "collect meat" task. Eureka iteratively employs an LLM (GPT-4o) to propose Python reward functions[3], in our case 5 options at each iteration, each of which is then used to train an RL agent (PPO) for $4 \times 10^5$ environment steps. Agent performance data is subsequently fed back to the LLM for generating improved reward functions. Eureka's objective was to design a reward function maximizing meat collection, and the RL agent trained *exclusively* on this LLM-generated reward signal, replacing the environment's intrinsic reward.

**Summary of Eureka Iterations and Results.** The primary metric for Eureka's success was the average reward over 10 trajectories achieved by the agent using the iteration's best reward function.

In the *Crafter Easy* (5 iterations), Eureka's LLM initial's attempt focused on food change and grassland exploration yielding –0.80 average reward. Subsequent iterations, reacting to feedback on ineffective or inconsistent rewards, saw the LLM emphasize meat collection (–0.45 average reward), incorporate sapling-to-cow breeding (0.60 average reward), refine temperature parameters (–0.25 average reward), and a final iteration resulted in 0.30 average reward.

In the *Crafter-Medium* (5 iterations), the initial reward design achieved –0.90 average reward. The LLM then focused on re-scaling penalties and applying exponential transformations, achieving –0.25 average reward. Further iterations attempted to reduce oversized food rewards by lowering temperature parameters (–0.65 average reward) and later discarded inactive components to concentrate on transformed food change rewards and moderated penalties (–0.75 average reward) before a final iteration that resulted in -0.9 average reward.

**Distinction from the Strategist Framework.** Eureka's results, while informative for reward design, are not directly comparable to the Strategist's due to fundamental differences in the goals of the two methods:

- **Role of LLM**: In the Strategist, the LLM is a high-level planner providing strategic guidance. In Eureka, the LLM is a reward function coder.

- **Core Objective**: The Strategist uses LLMs for top-down strategic planning, generating *shaping rewards* to guide an agent through a strategy. Eureka's aim is to *learn* the shaped reward itself, requiring multiple training iterations, and it does not aim for overall sample efficiency.

Note that each iteration takes 2 million environment interactions (aggregating the 5 options), and even if we consider just the interaction budget a single run in each iteration, average reward lags behind strategist-guided agents.

In essence, the Strategist enhances learning from existing environment signals via strategic context, whereas Eureka focuses on creating entirely new reward signals. These represent distinct research goals. The Eureka experiments confirm LLMs can iteratively generate reward code that leads to task achievement, but not necessarily improve sample efficiency given a new environment.

---

[3]We provide an equivalent context about the environment as to the Strategist and the Reward Shaper.

# D. Extended Related Work

*Table 11.* Comparison with Related Works. "Many Options" indicates whether the method proposes *multiple* policies or strategic paths rather than a single plan. "External Knowledge" indicates whether the method incorporates *external* knowledge, e.g., from LLMs or domain experts. "Flexible Primitives" indicates *flexible primitives* beyond a fixed set of macro-actions or text commands. "Multi-Level Abstraction" indicates whether it supports *multiple levels* of abstraction or only a single planning layer.

| Framework | Many Options | External Knowledge | Flexible Primitives | Multi-Level Abstraction |
|---|---|---|---|---|
| **Baseline Methods** | | | | |
| PPO (Schulman et al., 2017) | ✗ | ✗ | ✓ | ✗ |
| DreamerV3 (Hafner et al., 2023) | ✓ | ✗ | ✓ | ✗ |
| **Hierarchical RL** | | | | |
| SayCan (Ahn et al., 2022) | ✓ | ✓ | ✗ | ✗ |
| **Curriculum Learning** | | | | |
| Voyager (Wang et al., 2023) | ✗ | ✓ | ✓ | ✓ |
| **LLM-Based RL** | | | | |
| Yan et al. (Yan et al., 2024) | ✗ | ✓ | ✗ | ✗ |
| Zhang et al. (Zhang et al., 2024) | ✗ | ✓ | ✗ | ✗ |
| Liu et al. (Liu et al., 2024b) | ✗ | ✓ | ✗ | ✗ |
| ProgPrompt (Singh et al., 2022) | ✗ | ✓ | ✗ | ✗ |
| Lang. Planner (Huang et al., 2022) | ✓ | ✓ | ✗ | ✗ |
| Vid-Lang. Plan. (Du et al., 2024) | ✓ | ✓ | ✗ | ✗ |
| **LLM-Based Reward Shaping** | | | | |
| Sarukkai et al. (Sarukkai et al., 2024) | ✗ | ✓ | ✗ | ✗ |
| Ma et al. (Ma et al., 2024) | ✗ | ✓ | ✗ | ✗ |
| Xie et al. (Xie et al., 2024) | ✗ | ✓ | ✗ | ✗ |
| **Strategist (Ours)** | ✓ | ✓ | ✓ | ✓ |

Real-world decision-making tasks often pose significant challenges for traditional RL methods. The immense state-action spaces, sparse rewards, and evolving dynamics make it difficult for bottom-up approaches to efficiently learn optimal policies. As a result, researchers have explored various strategies to improve RL agents by incorporating external knowledge, structuring learning processes, and leveraging the recent powerful language models. In this section, we provide an extended review of related work, focusing on methods that align closely with our proposed strategist framework and comparing them specifically to our *Strategist* agent. Table 11 provides an overview of how our work differs from existing works across different subfields of reinforcement learning.

## D.1. Limitations of Traditional Reinforcement Learning Methods

Traditional RL methods such as Proximal Policy Optimization (Schulman et al., 2017) and DreamerV3 (Hafner et al., 2023) have achieved notable successes in controlled environments with well-defined state and action spaces. However, they operate under the bottom-up framework, relying heavily on trial-and-error interactions with the environment to learn optimal policies. This approach can be impractical in complex real-world scenarios where exploration is costly or risky.

**Proximal Policy Optimization (PPO):** PPO (Schulman et al., 2017) is an on-policy algorithm known for its balance between performance and computational efficiency. It updates policies by maximizing a clipped surrogate objective function, which helps maintain stable learning. However, PPO does not incorporate external knowledge or strategic planning into the learning process. It relies solely on interactions with the environment, making it inefficient in tasks with sparse rewards or where exploration is costly.

**DreamerV3:** DreamerV3 (Hafner et al., 2023) extends model-based RL by learning a world model that predicts future states and rewards. This allows agents to plan by imagining future scenarios, leading to improved sample efficiency. Although DreamerV3 can handle larger state spaces better than pure model-free methods, it still operates within the bottom-up

framework and does not leverage external knowledge or strategic planning at multiple abstraction levels.

**Environment Shaping:** In practice, to get these methods to perform acceptably on non-trivial tasks, practitioners often employ environment shaping (Park et al., 2024), which involves modifying the environment to facilitate learning. This requires significant human effort and expertise, reducing the flexibility of RL and potentially leading to overestimation of the capabilities of the methods. More critically, not all environments lend themselves to effective shaping.

In contrast, our *Strategist* integrates high-level knowledge and strategic planning to guide policy optimization without requiring human supervision. Using LLMs to incorporate external knowledge and generate strategies at various levels of abstraction, the *Strategist* reduces reliance on extensive exploration, allowing more efficient learning in complex environments.

### D.2. Curriculum Learning

To overcome the limitations of traditional RL, researchers have explored methods that integrate external knowledge to structure the learning process. Curriculum learning involves training agents on a sequence of tasks that gradually increase in difficulty (Bengio et al., 2009; Portelas et al., 2020). This approach helps agents build on previously acquired skills to tackle more complex tasks over time.

Traditional curriculum learning typically focuses on designing a progression of tasks that guide the agent toward the target task. The curriculum can be designed manually or automatically based on certain criteria, such as the agent's performance. However, it does not fully account for incomplete knowledge that might require the exploration of alternative strategies. Additionally, curriculum learning methods usually involve a trade-off between the effort to define the set of potential tasks within the environment and the quality or amount of learning needed to generate good curricula.

**Voyager:** The Voyager agent (Wang et al., 2023) exemplifies curriculum learning by autonomously creating and solving tasks in the Minecraft environment. It leverages external knowledge from LLMs to generate meaningful tasks and uses program synthesis to execute actions. Although Voyager incorporates external knowledge and operates at different abstraction levels, it focuses on sequential task progression without explicitly considering multiple strategic options.

Our *Strategist* agent builds upon these ideas by not only structuring the learning process but also explicitly generating and evaluating multiple strategies. This allows for more flexible and adaptive decision making, as the agent can consider alternative approaches and select the one that offers the best trade-off between feasibility and expected value.

### D.3. Hierarchical Reinforcement Learning

Hierarchical Reinforcement Learning (HRL) aims to decompose complex tasks into subtasks, allowing agents to learn policies at different levels of abstraction. By hierarchically structuring the decision-making process, HRL can significantly reduce sample complexity and improve learning efficiency.

**Feudal Reinforcement Learning:** Feudal learning, pioneered by Dayan & Hinton (1992), is one of the foundational approaches in HRL. In feudal learning, the agent is organized as a hierarchy of "managers" and "sub-managers," where higher-level managers set goals or subgoals for their sub-managers. Each manager operates at its own level of abstraction, making decisions based on a subset of the environment's state. While this introduces hierarchical control, early implementations typically required manual design of the hierarchy and subgoals, limiting adaptability. More recent approaches like Feudal Networks (Vezhnevets et al., 2017) leverage deep learning to learn these hierarchies but often focus on a fixed structure rather than exploring diverse high-level strategies.

**MAXQ Value Function Decomposition:** The MAXQ framework (Dietterich, 2000) offers another approach by decomposing the value function itself into a hierarchy. It breaks down a task into subtasks and learns a value function for each, representing the expected reward for completing that subtask and then following the optimal policy for the parent task. This allows for structured value-based planning but often relies on a pre-defined task graph, which may not capture all potential strategies or adapt easily to new information.

**Option-Critic Framework:** The Option-Critic Framework (Bacon et al., 2017) provides a method for learning "options," or temporally extended actions, in an end-to-end manner. Options consist of an intra-option policy (how to act), a termination condition (when to stop), and an initiation set (where the option can start). The framework learns both the policies and termination conditions using policy gradient methods, allowing agents to discover useful temporal abstractions autonomously. However, these learned options are typically grounded in the agent's experience and may not capture high-level strategies

that require external knowledge or complex reasoning.

**LLM-Based HRL (e.g., SayCan):** The SayCan framework (Ahn et al., 2022) combines LLMs with RL to enable robots to interpret high-level instructions and generate action plans. It uses LLMs to suggest potential actions (sub-goals) and RL (often through pre-trained value functions) to evaluate their feasibility. While powerful, SayCan and similar approaches often focus on sequential planning, require a fixed set of high-level action primitives, and do not typically reason across multiple levels of abstraction or explore fundamentally different strategic pathways.

**How the Strategist Differs:** In contrast, our *Strategist* agent explicitly generates a hierarchical Strategy Tree that considers various *approaches* (mutually exclusive strategies) and *plans* (sequences of sub-goals) at multiple levels of abstraction, *without* relying on a fixed set of primitives. Using LLMs, the *Strategist* incorporates external knowledge to generate and evaluate these different strategies based on estimated feasibility and expected value. This allows for a more comprehensive exploration of the *strategic landscape*. Unlike Feudal RL or MAXQ, which often require manual hierarchy/task design, or Option-Critic, which learns temporal abstractions from experience, the *Strategist* uses external knowledge (via LLMs) to *propose* and *evaluate* high-level strategic structures *before* committing to low-level learning. This enables it to consider complex, potentially counter-intuitive strategies that might not be discoverable through purely bottom-up or experience-driven HRL methods alone. While classic HRL methods focus on *executing* a (often pre-defined or learned) hierarchy, our framework emphasizes *generating* and *selecting* from *multiple* hierarchies (strategies), making it complementary; the *Strategist* could potentially propose high-level structures that classic HRL methods then learn to execute.

### D.4. Leveraging Large Language Models in Reinforcement Learning

Recent advances in LLMs have opened new avenues for integrating language understanding and reasoning into RL agents.

**Programmatic Prompting (ProgPrompt):** ProgPrompt (Singh et al., 2022) uses LLMs to generate code-like action plans for robots. By translating high-level instructions into executable programs, agents can perform complex tasks more effectively. However, ProgPrompt focuses on translating instructions into actions without considering alternative strategies and hierarchical levels of abstraction. Furthermore, it is only applicable in shaped environments that can be readily acted on using code.

**Language Planner:** The Language Planner (Huang et al., 2022) uses LLMs to generate action sequences for embodied agents in interactive environments. Although it leverages language understanding to produce plans, it does not generate these plans hierarchically at different levels of abstraction and is constrained to plan on a fixed set of primitives, limiting the reasoning flexibility that language provides.

**Video-Language Planning:** Video-Language Planning (Du et al., 2024) combines an LLM to plan, a generative diffusion video model as a world model, and a VLM to evaluate the resulting state predicted by the world model. However, this method does not incorporate high-level strategic planning and requires a practically omniscient world model that might be unavailable in many settings.

Our *Strategist* agent distinguishes itself by using LLMs for the construction of a hierarchical strategy tree that aids reasoning by considering decisions at different levels of abstraction. Assessment and choice between different strategies further solidifies our framework for complex decision making.

### D.5. Reward modeling

A key component of the *Strategist* agent is design appropriate reward signals to guide RL agents towards specific high-level policies. In this section we review different existing approaches for reward modeling.

**LLM-based reward modeling** LLMs have provided new opportunities for reward shaping. **Text2Reward** (Xie et al., 2024) leverages LLMs to generate dense reward functions as executable programs from natural language goals. **Eureka** (Ma et al., 2024) performs evolutionary optimization over LLM-generated reward code to exceed human-engineered rewards. Sarukkai et al. (2024) uses LLMs to author progress estimation functions that are converted into count-based intrinsic rewards for efficient learning. Although these methods enable the creation of dense rewards from natural language goals similarly to the *Strategist*, they require significant trial-and-error through either RL training iterations or human feedback collection to refine the reward functions. In contrast, the *Strategist* designs strategy-specific reward functions zero-shot, without the need for extensive environment interaction or human feedback loops.

**Inverse Reinforcement Learning**    While RL aims to learn an optimal policy from interactions with an environment and a reward signal, inverse RL (Ng et al., 2000; Abbeel & Ng, 2004) instead aims to recover the reward signal based on observed traces from an optimal policy interacting with the environment. One motivation for inverse RL is that the reward function provides a compact representation of the incentives behind an existing expert behavior than the underlying expert policy itself, allowing to learn the expert policy more efficiently than via direct imitation learning. More recently, IRL has played a crucial role in reinforcement learning from human feedback (RLHF) (Christiano et al., 2017; Bai et al., 2022a), enabling the extraction of implicit reward functions from human preferences for LLM alignment. However, the reliance on expert demonstrations limits the applicability of inverse RL to environment where such expert demonstrations are available. In contrast, the reward shaping part of the *Strategist* agent aims to design reward signal for specific strategic options, for which expert demonstrations might not be available.

**Self-Supervised Reward Modeling**    Self-Supervised Reward Modeling aims at autonomously generating reward functions that guide agent behavior without human supervision. **Plan2Explore** (Sekar et al., 2020) facilitates intrinsic reward shaping by predicting future novelty, enhancing exploration efficiency. **VIP** (Ma et al., 2022) learns from large-scale human videos to provide dense and smooth reward functions for unseen robotic tasks. These approaches reduce reliance on handcrafted reward functions and primarily focus on general reward function design as a pretraining mechanism. In contrast, the *Strategist* designs strategy-specific reward functions zero-shot, ensuring that the reward explicitly guides the agent toward a particular strategy rather than being designed for general exploration or pertaining.

### D.6. Human-Inspiration in Strategic Planning: A Bridge Between Cognition and RL

While our framework does not aim to be a precise cognitive model of human decision-making, its *top-down* approach is *inspired* by research in cognitive science and neuroscience on how humans tackle complex problems. Humans excel at navigating novel and intricate situations by not just reacting but by *strategizing*—forming high-level plans, considering alternatives, and abstracting away unnecessary details. This contrasts with traditional bottom-up RL, which often requires vast exploration. By drawing inspiration from these human capabilities, we aim to build RL agents that are more sample-efficient, flexible, and capable of tackling real-world challenges.

**Hierarchical Planning and Goal Decomposition.**    A cornerstone of human cognition is the ability to decompose complex goals into a hierarchy of manageable sub-goals. Early theories, such as those by Miller et al. (1960), proposed that hierarchical structures are fundamental to human problem-solving, echoing insights on serial order (Lashley, 1951) and means-ends analysis (Newell & Simon, 1972). This is supported by empirical evidence showing that people naturally break down tasks like the Tower of Hanoi or cooking into smaller steps. Neuroscience research further corroborates this, linking the prefrontal cortex to the segmentation of behavior into nested tasks and subtasks (Botvinick, 2008). Hierarchical Reinforcement Learning (HRL) mirrors this, and studies suggest a mapping between HRL mechanisms and neural structures (Botvinick et al., 2009), indicating that the brain might employ reusable sub-goal routines. This principle directly motivates our Strategist agent's design, which features a top-level agent orchestrating lower-level policies to handle decomposed sub-goals.

**Strategy Formation and Evaluation.**    Humans rarely commit to a single plan without considering alternatives. We maintain an "adaptive toolbox" of strategies (Gigerenzer & Todd, 1999; Payne et al., 1993) and select among them based on context. This involves exploring various approaches and refining them over time, as seen even in children's strategy development (Siegler, 1996). Crucially, humans engage in metacognitive evaluation, assessing the potential outcomes and costs (e.g., cognitive effort) of different strategies before acting (Lieder & Griffiths, 2017). Neural evidence suggests we can simulate these strategies internally, using mechanisms like hippocampal replay to pre-play future action sequences (Pfeiffer & Foster, 2013). This capacity for generating and evaluating diverse, high-level policies is a key inspiration for our Strategist agent, which explicitly builds and explores multiple strategic pathways (Approach Nodes) before committing to a plan.

**Structural Inference and Abstraction.**    Humans efficiently navigate complex environments by inferring latent structures and forming abstractions (Gershman & Niv, 2010). We don't treat every situation as new; instead, we identify higher-level patterns, rules, or categories that allow us to organize experience and generalize solutions. This "structure learning" enables us to create abstract representations—like mental landmarks in navigation or reusable sub-problems in mathematics—that significantly reduce planning complexity (Gershman & Niv, 2010). This ability also underpins transfer learning, where abstract principles learned in one context are applied to new situations. Our Strategist agent is informed by this, as it reasons

over a space of possible *policies* (structures of behavior) and evaluates their abstract properties, rather than getting bogged down in low-level action details.

**Cognitive Maps and Mental Models.** Effective planning often relies on internal models of the world, often termed "cognitive maps" or "mental models". Tolman (1948)'s work showed that even rats form internal representations that allow for flexible navigation and planning. More generally, mental models allow humans to simulate scenarios ("what if I do X?") and anticipate consequences without direct action, a concept dating back to Craik (1943). Neuroscience supports this, showing hippocampal activity representing possible future paths (Pfeiffer & Foster, 2013), effectively allowing the brain to "project" futures onto its cognitive map. Prefrontal regions interact with these maps, much like a planner querying a model. Our Strategist agent, by building and exploring a Strategy Tree, is implicitly creating an internal abstracted model—a map of the policy space—to guide its exploration and decision-making. Future works could explore in more depth the usage of more explicit world models within the strategist framework.

**Bounded Rationality and Plan Simplification.** Despite our planning abilities, human cognition is resource-limited—we operate under "bounded rationality," as described by Simon (1956). We don't exhaustively optimize; instead, we use heuristics and simplified models to make planning tractable, often "satisficing" with good-enough solutions. This involves pruning decision trees by ignoring unlikely or highly negative paths (Huys et al., 2012) and often preferring simpler policies even if they might yield slightly lower rewards (Lai & Gershman, 2024). This human tendency towards rational simplification and balancing costs (effort) with benefits (rewards) tangentially resonates with our Strategy Problem's core idea: finding policies that balance *feasibility*, *specificity* (simplification, pruning) and *value* (optimality).

In summary, human planning is characterized by its hierarchical nature, its exploration of multiple strategies, its use of abstraction and internal models, and its resource-rational approach to simplification. By drawing inspiration from these cognitive hallmarks, our top-down RL framework and the Strategist agent aim to equip RL systems with a more efficient, structured, and human-like approach to navigating complex decision-making landscapes.

## D.7. A Real-World Illustrative Example: Hospital Management

To further illustrate the applicability and intuition behind our top-down strategic approach—especially in complex, real-world domains where bottom-up exploration is impractical due to vast state spaces, sparse rewards, and high-stakes decisions (Dulac-Arnold et al., 2021; Rengarajan et al., 2022)—we present an example based on hospital management. This example aims to provide a more broadly relatable illustration of how strategic decomposition, inspired by human planning (Correa et al., 2023; Collins et al., 2024), can address challenges where traditional RL struggles. Consider the goal of improving patient discharge rates, a common challenge in healthcare operations (Strickland & Lizotte, 2021).

**Real-world Example: Improving Discharge Rate in a Hospital**

**Environment: Real-world hospital** 🏥
**Goal: Maximize Discharge Rate** 🛏

**Bottom-up Approach**

*Impractical due to prohibitively large state-action space and limited margin for trial-and-error, where failures can have significant costs (Dulac-Arnold et al., 2021).*

**Top-down Approach**

- **Identify current bottlenecks**: E.g., insufficient community care, scheduling gaps, or prolonged recovery times (Strickland & Lizotte, 2021).

- **Derive strategic options**: E.g., streamline coordination with social services or improve recovery pathways, decomposing high-level goals (Correa et al., 2023).

- **Implement and monitor**: Select the most promising plan, implement it, and regularly review its effectiveness.

*Summary:* Bottom-up approaches often struggle in such open-ended, high-risk settings. A top-down approach intelligently diagnoses the initial problem, contemplates different plausible strategies, and focuses efforts on the most relevant ones, reducing the exploration burden much like human experts do (Sutton et al., 1999; Collins et al., 2024).

# E. Extended limitations and future work

This section expands on the primary constraints of our framework and outlines promising directions for future research. Although the Strategist agent represents a principled step toward incorporating high-level human-inspired reasoning into RL, each limitation highlights pathways for deeper improvements.

## E.1. Limitations

**Reliance on LLMs.** Our framework leverages LLMs to generate strategies, estimate feasibility and value, and assess specificity. While these models encode broad knowledge and can produce coherent strategies, they are susceptible to inaccuracies such as factual errors, hallucinations, or bias (Wei et al., 2025). LLMs may propose strategies that are infeasible or misaligned with real dynamics, which we try to palliate by introducing the notion of *feasibility*. Moreover, reliance on pre-trained models implies an assumption of sufficiently rich domain knowledge in the LLM's parameters, which may not always hold (Vafa et al., 2024). In domains where up-to-date or highly specialized information is unavailable, LLM-based strategies risk becoming misleading or incomplete.

**Accuracy of the Strategist's Metric Estimations.** Feasibility and value are core metrics for pruning the search space and identifying promising policies; however, their quality depends on LLM-based approximations, which our empirical results suggest are imperfect. Without an environment feedback loop, the Strategist currently has no mechanism to adapt feasibility/value estimates based on actual agent performance. Although promising but slightly inaccurate strategies may still accelerate exploration, genuinely flawed strategies—especially in safety-critical applications—could yield wasted computational effort or detrimental real-world outcomes.

**Integration with Low-Level Policies.** Bridging the gap between abstract strategies and concrete actions remains a challenge. Even if a strategy appears "feasible" at a high level (e.g., "collect saplings, then breed cows"), there is no guarantee an RL agent will be able implement the sub-goals. The coordination between top-down plans and bottom-up policy learning can break down if the strategy is misaligned with the agent's capabilities or environment constraints. Adaptive mechanisms for on-the-fly revisions in response to low-level agent feedback would be required to address this.

**Scalability and Latency.** Querying an LLM for sub-goal evaluations introduces overhead, particularly for large domains or frequent strategy updates. Although we reduce costs by distilling LLM-based scores into a learned reward-shaping model, computational constraints could remain a concern. In some settings, even moderate overhead could be prohibitive.

**Larger Evaluation Before Real-world Deployment.** Our current evaluations primarily demonstrate improvements in exploration and sample efficiency on moderately complex tasks. However, larger benchmarks that comprehensively capture sparse rewards, large state-action spaces, and safety constraints are not provided. More extensive or domain-specific evaluations (e.g., in robotics, supply chain optimization, or clinical decision support), would be required before advocating for real-world deployment.

**Ethical and Safety Considerations.** Real-world domains (e.g., healthcare, finance) impose strict ethical and legal requirements, making suboptimal exploration especially risky. While top-down strategies can reduce harmful exploration, they do not guarantee compliance with external regulations or ethical norms. LLM biases might also inadvertently introduce fairness or safety concerns (Wei et al., 2025). Integrating explicit safety checks or moral-legal filters into strategy generation is critical to ensure that the Strategist remains compliant when deployed beyond simulated environments.

## E.2. Future Work

**Real-World Feedback Loops.** To improve the reliability of feasibility and value estimations, one natural extension is to close the loop between the Strategist and the environment. After initial strategy selection, the agent's observed performance and environment feedback could update LLM-based metrics, prompting refined strategy suggestions or pruning. Such iterative, active-learning procedures would mitigate errors arising from purely static, zero-shot LLM predictions. It would also be interesting to investigate how a strategist can be used to pivot from an incumbent policy.

**Enhancing LLM Capabilities and Bias Mitigation.** Fine-tuning or instruction-tuning LLMs on domain-specific data might reduce inaccuracies. In parallel, systematic bias detection and mitigation (with techniques such as Constitutional AI

(Bai et al., 2022b)) are crucial if top-down strategies are deployed in sensitive domains where fairness is paramount.

**On-the-Fly Re-Strategizing.** Real-world tasks often present non-stationary conditions, unforeseen obstacles, or shifting resource constraints. An important open problem is enabling the Strategist to revise or pivot among multiple high-level strategies mid-training, rather than committing to a single final plan. Bayesian updates to feasibility and value, or active assessment of partial sub-goals, could enable dynamically selection among promising strategies.

**Top-Down vs. Bottom-Up Synergies.** While our experiments emphasize top-down guidance, purely bottom-up RL retains advantages in discovering unanticipated solutions or optimizing fine-grained control, which underpins our use of a decaying weight for the shaped reward. Future work could investigate further hybrid frameworks that initially constrain exploration through top-down plans, then loosen these constraints to allow bottom-up refinements. Such synergies might prove valuable in dynamic tasks where specifying every sub-goal a priori is infeasible. Future works could also investigate how to combine the strategist with other bottom-up methods such as Q-learning (Li et al., 2024), among others.

**Incorporation of action-biases.** As explained on the main text, works such as, (Yan et al., 2024), (Zhang et al., 2024) or (Liu et al., 2024b) use LLMs to bias actions. These methods are orthogonal to ours, and future work could be investigate using them in tandem.

**Safe, Multi-Agent, and Continuous Deployment.** Beyond single-agent settings, extending the Strategist to multi-agent coordination, competitive games, or collaborative tasks would necessitate strategy-level negotiation among agents. In continuous deployment scenarios (e.g., warehouse robotics or healthcare scheduling), strategies may need periodic re-synthesis to remain optimal under evolving conditions. Additionally, explicit safety and ethical constraints must be baked into both the strategy-generation mechanism (LLM queries and reasoning) and the learned sub-policies to ensure responsible real-world usage.

