# OpenReview forum: "Strategic Planning: A Top-Down Approach to Option Generation"
_ICML.cc/2025/Conference — ICML 2025 poster_

### Official Review · Reviewer_JCuE · 2025-03-08

**Overall Recommendation:** 3

**Summary:**

This paper proposes a top-down learning method for Reinforcement Learning (RL). The method leverages a Large Language Model (LLM) to decompose a complex task from high-level goals into fine-grained plans, considering specificity, value and feasibility. Following decomposition, these plans are transformed into reward signals through reward shaping for RL training. The paper demonstrates the effectiveness of using rewards derived from LLM decomposition. In experiments, the proposed method is evaluated on the Crafter environment, and the results show that it outperforms conventional RL methods.

**Claims And Evidence:**

This paper claims that bottom-up approaches learn low-level actions and improve through extensive trial and error, and argues that LLM-augmented RL methods (e.g., Hierarchical RL, LLM-as-policy, Reward shaping) also inherit the drawbacks associated with the bottom-up paradigm. However, these methods can also accelerate the convergence of RL agents to optimal solutions by leveraging language priors, which reduce the need for large-scale trial and error. Consequently, the primary advantage of top-down approaches remains unclear, and the paper lacks experimental comparisons between its proposed method and other LLM-augmented RL techniques.

**Essential References Not Discussed:**

None.

**Experimental Designs Or Analyses:**

The baselines are lacking. In Sections 1-3, this paper discusses the advantages of top-down approaches compared to bottom-up ones that are also augmented by LLMs (e.g., LLM as policy, hierarchical RL). However, in the experiments, the authors only compared their method with traditional RL approaches, such as PPO, and did not conduct comparisons with bottom-up approaches that also use LLMs. This omission makes it difficult to demonstrate the superiority of the top-down approach in scenarios where LLMs are introduced in both paradigms.

**Methods And Evaluation Criteria:**

The evaluation benchmark is limited, consisting of only two tasks, "Hunt" and "Breed," which are not sufficiently representative.

**Other Comments Or Suggestions:**

None.

**Other Strengths And Weaknesses:**

** Strengths**

1. This paper introduces a top-down paradigm for RL with LLMs, offering a novel perspective on task decomposition and RL learning.

2. The paper proposes a flexible tree structure for task decomposition, featuring two node types – approach nodes and plan nodes – and three selection metrics.

**Questions For Authors:**

1. In Section 4.3, how is the entropy H calculated using LLM?

2. In Section 4.3, how is the value v calculated? Is J(·) obtained through rollout or from the environment’s reward process?

3. In Section 4.3, what does the probability in feasibility refer to?

4. In lines 152-153, it is stated that the entropy H needs to be lower than $H_{max}$ to satisfy being sufficiently specific. In Section 4.3, how does the calculation of specificity ensure this requirement is met?

**Relation To Broader Scientific Literature:**

This paper enhances RL learning using an LLM following a top-down paradigm. While previous research has also explored LLM-augmented RL, these studies typically adopt a bottom-up approach, which often requires extensive trial-and-error.

**Theoretical Claims:**

I have checked theorem 2.1.

---

> ### Author Rebuttal · Authors · 2025-04-01
>
> We thank the reviewer for their insightful review and for recognizing the novel perspective.
>
> We appreciate the reviewer's concern about lacking comparisons between our top-down approach and bottom-up approaches that also leverage LLMs. This is a valid point, and we have now expanded our evaluation to include **two additional strong RL baselines (DreamerV3 [1] and EDE [2])** as well as **two LLM-based bottom-up approaches (LLM-as-policy and Eureka [3])** to provide a more comprehensive assessment of our method's effectiveness.
>
> ## Extension to other RL baselines
> We've added **DreamerV3** (50M parameters) [1] and **Exploration via Distributional Ensemble (EDE)** [2], both highly ranked on the original Crafter leaderboard. **DreamerV3** achieves slightly better performance than PPO (0.15 ± 1.39, N=4) but remains below our Strategist + PPO framework. [Preliminary results](https://imgur.com/a/NuBI80H) indicate that **EDE** reaches high reward (~6 avg after 2M training steps) by initially discovering a hunting strategy and subsequently learning survival skills to lengthen episodes. Critically, our Strategist framework can be combined with EDE, enabling it to be steered toward distinct strategies such as hunting and breeding. While the hunting strategy performs comparably to early EDE stages, the breeding strategy—which was overlooked by vanilla EDE—enables a **2× reward improvement over EDE's final performance**. This demonstrates that our approach is broadly applicable across different RL algorithms and can enhance the exploration capabilities of existing methods.
>
> ## New LLM-based baseline experiments
> We implemented an **LLM-as-policy** approach (shown to be effective in prior works such as Voyager and Reflexion), using GPT-4o-mini for direct action selection at each timestep. Notably, this is the same model used to annotate states in the Strategist’s reward shaping procedure to ensure a consistent LLM backbone across both baselines and our main method. This approach achieved 0.43 ± 1.01 reward (N=15), substantially underperforming our Strategist framework combined with PPO or EDE agents while being computationally expensive due to per-step inference requirements. Additionally, we evaluated **Eureka** [3] with GPT-4o, a state-of-the-art bottom-up LLM-based iterative reward discovery method. In our preliminary experiments with an evolutionary search (population=3, depth=3), Eureka's designed rewards didn't meaningfully outperform the environment reward, yielding results equivalent to baseline PPO while requiring a higher exploration budget. These comparisons directly demonstrate the advantages of our top-down strategic approach over bottom-up LLM-augmented alternatives in terms of both effectiveness and computational efficiency.
>
> ## Distinction between bottom-up and top-down methods
> We appreciate the clarification requested regarding the distinction between top-down and bottom-up methods. Indeed, we agree that bottom-up approaches augmented with LLMs can significantly improve sample complexity, and we emphasize that our top-down method is not inherently mutually exclusive but rather orthogonal to these bottom-up methods. Consequently, direct empirical comparisons were initially omitted as our goal was not to position these approaches in competition, but to present a complementary framework. For instance, methods using an LLM as a policy or hierarchical RL can be readily integrated alongside our Strategist framework to potentially further improve sample efficiency, as discussed in our future work section.
>
> ## Additional comments and revisions
> We acknowledge the importance of the detailed questions raised about Section 4.3, and will address these clearly in a newly added appendix section, briefly summarized here due to character constraints:
>
> - Regarding feasibility, this probability represents how much of the probability mass of the current strategy aligns with policies achievable by the agent’s learning hypothesis class.
> - Initially, attributes like specificity, feasibility, and value are estimated by the LLM based on its prior knowledge. As demonstrated in Section 5.3, these estimates can be dynamically updated with actual environment interactions, opening possibilities for future multi-armed bandit-inspired strategy adjustments during learning, which we encourage in our future work discussion.
>
> Thank you again for your valuable insights and constructive suggestions.
>
> ## References
> [1] Hafner, Danijar, et al. "Mastering diverse domains through world models.".
>
> [2] Jiang, Yiding, J. Zico Kolter, and Roberta Raileanu. "Uncertainty-driven exploration for generalization in reinforcement learning.".
>
> [3] Ma, Yecheng Jason, et al. "Eureka: Human-Level Reward Design via Coding Large Language Models.".

---

### Official Review · Reviewer_rDHf · 2025-03-10

**Overall Recommendation:** 2

**Summary:**

The paper proposed Strategic Planning, a top-down approach for decomposing complicated reinforcement learning (RL) task into natural language-described sub-tasks. The paper also designs a reward shaping methodology that translates these strategies expressed in natural language into quantitative feedback for RL methods. Detailed implementation is introduced with theoretical justifications. The overall pipeline has been tested on Crafter environment to show its effectiveness.

**Claims And Evidence:**

See below.

**Essential References Not Discussed:**

Many classic hierarchical RL studies should be discussed. E.g., Feudal RL actually shares similar high-level idea of top-down hierarchy.


- Dayan P, Hinton G E. Feudal reinforcement learning[J]. Advances in neural information processing systems, 1992, 5.

- Dietterich T G. Hierarchical reinforcement learning with the MAXQ value function decomposition[J]. Journal of artificial intelligence research, 2000, 13: 227-303.

- Vezhnevets A S, Osindero S, Schaul T, et al. Feudal networks for hierarchical reinforcement learning[C]//International conference on machine learning. PMLR, 2017: 3540-3549.

**Experimental Designs Or Analyses:**

Yes, I have checked. See below.

**Methods And Evaluation Criteria:**

Yes, it makes sense.

**Other Comments Or Suggestions:**

- Figrue 4 contains an 80% percentile confidence interval, which is not a common choice (95% or 68% is more common), is there any reason to do so?
- The Crafter environment was modified by the authors in this study? Could the authors explain why? Can Strategic Planning also work in the non-modified version?

**Other Strengths And Weaknesses:**

### [Strengths]
- Ideas are theoretically justified.
- Figures are nicely made.
- The writting is easy to follow.


### [Weakness]
- The largest concern I've got is the to what degree the proposed method can be effective in general. If I understood correcly, Strategic Planning requires humans to input a textual strategy, e.g., "collect much meat as much as possible". I am not sure how much this can generalize to other tasks. How do you know a strategy is good in general?
- The comprehensiveness of empirical results is yet to be improved. That is to say, it is expected to have more types of tasks and environments to be evaluated, and more strong baselines (rather than vanilla PPO, but those also used LLMs, i.e., SayCan and Eureka) to be compared with, to meet the quality bar of ICML.
- Using Minecraft-Collecting Meat as an illustrative example may not be a proper choice since not everyone is familier with it.
- The introduction is broken into several parts with respective subtitles. This may prevent the reading flow and make readers feel discountinual.
- Many classic works on hierarchical RL are ignored, see "Essential References Not Discussed".

**Questions For Authors:**

See above.

**Relation To Broader Scientific Literature:**

The current paper discussed using LLM and human prompts to make more effective hierarchy of planning and design auxillary reward functions to help solving RL problems. The high-level idea of using LLM to make plans and design options are not new (e.g., SayCan), the using LLM to design reward functions is not new (e.g., Eureka). However, the current paper is novel in terms its detailed implementation and methodology.

**Theoretical Claims:**

No problem with me.

---

> ### Author Rebuttal · Authors · 2025-04-01
>
> We sincerely thank you for your detailed and insightful feedback.
>
> ## Additional RL baselines
> Following your suggestion, we have added **two competitive RL baselines**: DreamerV3 [1] and Exploration via Distributional Ensemble (EDE) [2], both high ranking on the original Crafter leaderboard. DreamerV3 achieves slightly better performance than PPO (0.15 ± 1.39 (std, N=4)) but remains below our Strategist + PPO framework.
>
> [Preliminary results](https://imgur.com/a/NuBI80H) indicate that EDE reaches high reward on Crafter-Medium (~6 avg) by initially discovering a hunting strategy and subsequently learning survival skills to lengthen episodes further. However, our Strategist approach is orthogonal to the underlying RL agent: combined with EDE, the Breeding strategy—which vanilla EDE does not exploit—results in a **2× increase in rewards compared to EDE’s final performance**. This highlights the value of our top-down approach in uncovering effective, alternative strategies that traditional bottom-up methods may not discover.
>
> ## LLM baselines
>
> We agree it is important to compare our method to other LLM-based baselines. We have added **two additional LLM baselines**: LLM-as-Policy and Eureka [3].
>
> First, we implemented an **LLM-as-Policy** baseline with GPT-4o-mini, which selects actions at each step directly. This baseline achieved only 0.43 ± 1.01 reward (N=15), significantly underperforming compared to our Strategist framework. Furthermore, its computational cost is high due to requiring inference at each timestep.
>
> Second, we compared our method to **Eureka**, an LLM-based approach designed for iterative reward discovery using GPT-4o. Eureka’s primary objective is finding high-quality reward functions, without aiming for sample efficiency. In our experiments, Eureka's evolutionary search (population size 3, depth 3) did not yield rewards meaningfully superior to the default environment reward, performing equivalently to baseline PPO despite utilizing greater exploration resources. For completeness, we will provide results with a deeper Eureka run in the revised manuscript.
>
> SayCan is designed for robotic tasks using predefined action primitives and a pretrained value function for feasibility evaluation. This setup does not transfer well to environments where the space of meaningful behaviors is broader and the choice of primitives is not natural, plus the existence of a pre-trained low-level actor is a strong requirement, rarely available. Instead, our method leverages LLMs for high-level strategy generation and uses lang. cond. RL to ground those strategies through interaction. In practice, approaches like SayCan and action-proposing LLMs are actually orthogonal to ours—they focus on low-level decision-making given a set of primitives, which could be combined with our high-level strategic planning and exploration.
>
> ## Additional comments
> - **Generalizability**: we clarify that one of the strategist's strengths is precisely its flexibility. Our method is environment-agnostic conditioned on the LLM being able to generate meaningful strategies given domain descriptions, which is reasonable for a broad range of tasks. Unlike other methods, the Strategist does not require pre-training, expert demonstrations or predefined sub-goals.
> - **Hierarchical RL**: Thank you for pointing out key prior work. We expanded Appendix D.3 to better situate our method within the hierarchical RL literature. In addition to citing Feudal RL (Dayan & Hinton, 1992), we now include discussions of MAXQ (Dietterich, 2000) and Feudal Networks (Vezhnevets et al., 2017), highlighting both differences and potential complementarities with our framework.
> - **Confidence intervals in Fig. 4**: We acknowledge the nonstandard use of the 80% interval. We now include 68% intervals in the appendix, noting that this change does not affect the qualitative conclusions.
> - **Crafter environment modifications**: We modified Crafter to better expose strategic trade-offs (e.g., hunting vs. breeding), making the evaluation of strategy discovery more explicit. Our method remains applicable to the original environment, but we found it less effective for highlighting the core contributions given its achievement-based reward.
> - **Minecraft-meat collection example**: To address accessibility, we added a real-world motivating example (hospital management) in the appendix, illustrating how our framework supports discovering diverse high-level strategies in complex decision-making settings.
>
> Thank you once again for your constructive critique and valuable recommendations.
>
> ## References
> [1] Hafner, Danijar, et al. "Mastering diverse domains through world models.".
>
> [2] Jiang, Yiding, J. Zico Kolter, and Roberta Raileanu. "Uncertainty-driven exploration for generalization in reinforcement learning.".
>
> [3] Ma, Yecheng Jason, et al. "Eureka: Human-level reward design via coding large language models."

---

> > ### Comment · Reviewer_rDHf · 2025-04-02
> >
> > I appreciate the authors' response and conducting comparison with more alternative methods. While some part of my concern has been mitigated, a major concern remains that how the proposed method works in diverse tasks. I updated my score to 2 accordingly.

---

> > > ### Author Response · Authors · 2025-04-07
> > >
> > > Thank you for your engagement and for acknowledging our additional experiments and comparisons. We appreciate the opportunity to further clarify your concerns. Below, we address them in two parts: **(1) why our current experimental setting already demonstrates general applicability**, and **(2) we present new results on the original Crafter environment that further confirm the broader potential of our approach**.
> > >
> > > ---
> > > # Why Current Experimental Setting Demonstrates Broader Potential
> > > ## 1. General Conditions for Applicability
> > > Our method requires only two general conditions:
> > > - A capable LLM that can propose meaningful domain-relevant strategies.
> > > - An ability to interpret environment states into signals aligned with these strategies.
> > >
> > > We explicitly discuss in the manuscript that if an LLM fails to generate relevant strategies or to interpret observations adequately, our method naturally becomes ineffective. However, modern LLMs already exhibit strong reasoning capabilities and broad domain coverage, implying our framework is readily applicable to a diverse set of real-world tasks.
> > > ## 2. Insights Provided by Our Modified Crafter Tasks
> > > We deliberately designed our modified Crafter tasks to test multiple strategic dimensions, including short-horizon (hunting) and long-horizon (breeding) objectives, across two difficulty levels (Easy and Medium). This experimental design explicitly evaluates our Strategist’s capacity to guide agents toward complex, delayed-reward strategies that even state-of-the-art methods (e.g., DreamerV3, EDE) fail to uncover. Specifically, our experiments demonstrate:
> > > 1. **Superior Performance:** Strategist-guided agents consistently outperform the baselines.
> > > 2. **Strategy-Driven Exploration:** The Strategist directs the agent toward approaches that would otherwise remain unvisited. For instance, combining our method with EDE unlocked breeding behaviors EDE alone never found.
> > > 3. **Consistency Check:** We verify that the LLM’s preliminary feasibility/value estimates correlate with final RL outcomes.
> > >
> > > Together, 1 - 3 provide confidence that the improvements are neither spurious nor narrowly overfitted. Thus, our setup already indicates that the Strategist can (i) learn complex behaviors from strategies, (ii) significantly reduce reliance on lucky exploration, and (iii) integrate smoothly with existing RL algorithms.
> > >
> > > ---
> > > # Additional Experiments in the Original Crafter Environment
> > > To further confirm the generality of our approach, we evaluated Strategist+EDE on the (official) Crafter environment, which includes built-in achievement-based rewards. The Strategist proposed four distinct approaches to collecting achievements—Fight, Craft, Resources, and Needs—each prioritizing a subset of achievements (see the center top panel in the results, linked below). We then compared each to vanilla EDE. These preliminary results are promising and clearly illustrate our method’s broad applicability. Full results are available [here](https://imgur.com/a/fTj7F2G).
> > > 1. **Overall Achievement Gains (Fig.1):**
> > >    Strategies focused on Fighting, Crafting, and Resources outperform baseline EDE by approximately +1.58, +1.48, and +1.35 achievements per episode, respectively (baseline average: 7.48).
> > >    - The Needs strategy performs similarly to baseline, since survival-related achievements are easily reached through exploration.
> > > 2. **Strategy-Specific Milestones (Fig.2):**
> > >    - Each strategy uniquely excels in tasks aligned with its intended focus:
> > >      - **Fight** significantly accelerates achievements related to weapon crafting (stone swords) and combat tasks (e.g., defeating skeletons).
> > >      - **Craft** builds stone swords and pickaxes, and together with **Resources**, they are the only ones to succeed in advanced resource collection tasks (e.g., gathering iron, see Fig. 3), all of which are undiscovered by baseline EDE.
> > >    - Figures 2 illustrate these gains clearly, demonstrating rapid achievement of specialized goals that baseline methods rarely or never discover.
> > > 3. **Distinct Strategy Footprints (Fig.3):**
> > >    - The heatmap reveals distinct and interpretable patterns of achievement completion that uniquely characterize each strategy, clearly emphasizing the specific tasks each was designed to target. Anecdotally, when provided with this table with blanked-out column names, an o1 model can accurately infer which achievement completion profiles correspond to each strategy, highlighting the clarity and effectiveness of policy steering.
> > >
> > > Collectively, these results demonstrate that our Strategist framework generalizes effectively beyond the initial domain modifications and across multiple tasks. By validating our method on an additional, widely used benchmark task using more open-ended strategies, we believe this directly addresses your concern regarding general applicability.
> > >
> > > ---
> > > Thank you again for your feedback and valuable suggestions.

---

### Official Review · Reviewer_c4E9 · 2025-03-13

**Overall Recommendation:** 3

**Summary:**

The paper considers a top-down approach to hierarchical planning/option generation. The proposed Strategist Agent builds a tree structure that specifies alternative plans or sequential plans (approach and plan nodes), which are broken further, if necessary. The tree structure is generated by a sufficiently `strong' LLM with prompts describing the design of the Strategist Agent, and a sufficiently rich description of the domain. The LLM attaches a reward function and a feasibility evaluation to the nodes as well.

The proposed algorithm is illustrated on the Crafter environment. There is an empirical comparison showing favorable performance compared to PPO.

The paper also include a generic result that links the specificity and value of a policy to the sample complexity.

**Claims And Evidence:**

The paper claims empirical superiority over PPO on the task considered, which is not surprising for a task with sparse rewards and long scenarios. The approach seems interesting enough, but the evaluation is limited.

**Essential References Not Discussed:**

The most relevant work is sufficiently discussed (especially in the supplementary material)

**Experimental Designs Or Analyses:**

I have looked closely to the empirical design, including the prompt details.

**Methods And Evaluation Criteria:**

The evaluation method is reasonable, but the set of baselines is very limited. Even in the absence of a larger pool of implemented/tested baselines (there should have been some that are suited to deal with sparse rewards and long episodes), the test environment could have been chosen such that the scoreboard of Crafter would have been a measure for comparison.

**Other Comments Or Suggestions:**

The authors argue that SayCan can is more limited because the options are specified bottom-up, while here it is completely top-down. While this is true, I would add that the two operated in a very different environments: the environment discussed here is essential a strategy game, while SayCan deals with robots. It is much easier to evaluate the feasibility and shape reward function in the former. LLMs would probably would have more difficulty in defining options in the latter. The prompts detailed in this paper are also quite rich in domain knowledge that helps defining the rewards.

**Other Strengths And Weaknesses:**

The prompts and the generated plans are quite illustrative.

**Questions For Authors:**

Would it be possible to compare Strategist Agent with the leader board for Reacher?

**Relation To Broader Scientific Literature:**

It is an interesting approach, very much relevant for training hierarchical agents using LLM.

**Theoretical Claims:**

I parsed the proofs, but the theoretical result has a minor role in the paper anyway.

---

> ### Author Rebuttal · Authors · 2025-04-01
>
> We appreciate the reviewer's feedback regarding our evaluation methodology. To address these concerns, we have substantially expanded our experimental comparisons to include **two additional RL baselines specifically designed for sparse reward settings and long episodes (DreamerV3 [1], EDE [2]), as well as two LLM-based approaches (LLM-as-policy, Eureka [3]).**
>
> ## Model-based and exploration-focused RL baselines
> We now include **DreamerV3** (50M parameters) [1], a leading model-based RL approach, which slightly outperforms PPO but still falls short of our Strategist+PPO combination. Additionally, we evaluate **EDE** (Exploration via Distributional Ensemble) [2], another high-performing method on the original Crafter leaderboard. [Preliminary results](https://imgur.com/a/NuBI80H) indicate that it achieves strong performance on Crafter-Medium (~6 avg reward after 2M training steps) by following a hunting strategy while learning to maximise survival to extend episodes. Notably, our Strategist framework is complementary to the underlying RL algorithm. When paired with EDE, it introduces a breeding strategy that vanilla EDE fails to uncover. This results in a **2× improvement in reward**, highlighting how high-level strategic guidance enhances and expands the exploration capabilities of existing methods. This also demonstrates that our method is broadly applicable across different RL approaches and can draw from the strengths of each method depending on the environment of interest.
>
> ## Comparison with LLM-based approaches
> We have expanded our evaluation to include two prominent LLM-based baselines to clarify our method's advantages. First, we implemented an **LLM-as-Policy** baseline using GPT-4o-mini, which selects actions at each step directly. This baseline achieved only 0.43 ± 1.01 reward (N=15), significantly underperforming compared to our Strategist framework. Furthermore, its computational cost is high due to requiring inference at each timestep, which underscores its impracticality for complex environments.
>
> Second, we compared our method to **Eureka** [3], an LLM-based approach designed for iterative reward discovery using GPT-4o. Eureka’s primary objective is finding high-quality reward functions, which often comes at the expense of sample efficiency. In our experiments, Eureka's evolutionary search (population size 3, depth 3) did not yield rewards meaningfully superior to the default environment reward, performing equivalently to baseline PPO despite utilizing greater exploration resources. For completeness, we will also provide results with a deeper Eureka run in the revised manuscript. These results clearly illustrate that our Strategist framework—translating high-level LLM-generated strategies into actionable rewards—is significantly more effective and sample-efficient than both direct LLM action-selection and standard iterative reward optimization methods, especially in complex, strategic domains.
>
> ## Modified Crafter environment
> Regarding the modification of the Crafter environment, our intention was to explicitly showcase the benefits of strategic exploration facilitated by our approach. By adjusting the environment to feature clear alternative strategies, such as hunting versus breeding, we aimed to rigorously test the Strategist Agent's ability to identify and exploit distinct high-level strategies effectively.
>
> ## Comparison with SayCan
> We thank the reviewer for highlighting important differences between our method and SayCan. Indeed, SayCan addresses bottom-up planning in robotic environments, which typically admit defined primitive actions. In such contexts, feasibility assessment through pretrained value functions is straightforward but requires costly pre-training of low-level actors, limiting sample efficiency.
>
> Conversely, our top-down Strategist Agent is tailored for complex strategy-driven environments that inherently have a wide range of possible high-level actions. Unlike SayCan, our method does not rely on a fixed set of predefined primitives, making it suitable for broader strategic exploration. Our approach fundamentally capitalizes on the domain-agnostic strategic planning capabilities of LLMs combined with language-conditioned RL, enabling iterative refinement and robust exploration. Thus, our framework is uniquely positioned to effectively address settings where bottom-up methods like SayCan would be less feasible or impractical.
>
> Thank you again for your valuable feedback and suggestions.
>
> ## References
> [1] Hafner, Danijar, et al. "Mastering diverse domains through world models.".
>
> [2] Jiang, Yiding, J. Zico Kolter, and Roberta Raileanu. "Uncertainty-driven exploration for generalization in reinforcement learning.".
>
> [3] Ma, Yecheng Jason, et al. "Eureka: Human-Level Reward Design via Coding Large Language Models.".

---

> > ### Comment · Reviewer_c4E9 · 2025-04-03
> >
> > The additional experiments do indeed improve the paper.

---

> > > ### Author Response · Authors · 2025-04-07
> > >
> > > Thank you for acknowledging that our additional experiments with DreamerV3, EDE, and LLM-based baselines have improved the paper. We are glad these comparisons addressed many of your prior concerns.
> > >
> > > To further confirm the generality of our approach, we also conducted **additional experiments** evaluating **Strategist+EDE** on the **original** (official) Crafter environment, which includes built-in achievement-based rewards. The Strategist proposed four distinct approaches to collecting achievements—Fight, Craft, Resources, and Needs (see the center top panel in the results, linked below)—each prioritizing a subset of achievements. We then compared each to vanilla EDE. Full results are available [**here**](https://imgur.com/a/fTj7F2G). We briefly summarize these compelling new results below, further highlighting the flexibility and general applicability of our Strategist framework:
> > >
> > > ### Additional Experiments in the Original Crafter Environment
> > > 1. **Overall Achievement Gains (Fig. 1):**
> > >    Strategies focused on Fighting, Crafting, and Resources outperform baseline EDE by approximately +1.58, +1.48, and +1.35 achievements per episode, respectively (baseline average: 7.48).
> > >    - The Needs strategy performs similarly to baseline, since survival-related achievements are easily reached through exploration.
> > > 2. **Strategy-Specific Milestones (Fig. 2):**
> > >    - Each strategy uniquely excels in tasks aligned with its intended focus:
> > >      - **Fight** significantly accelerates achievements related to weapon crafting (stone swords) and combat tasks (e.g., defeating skeletons).
> > >      - **Craft** builds stone swords and pickaxes, and together with **Resources**, they are the only ones to succeed in advanced resource collection tasks (e.g., gathering iron, see Fig. 3), all of which are undiscovered by baseline EDE.
> > >    - Figure 2 illustrate these gains clearly, demonstrating rapid achievement of specialized goals that baseline methods rarely or never discover.
> > > 3. **Distinct Strategy Footprints (Fig. 3):**
> > >    - The heatmap reveals distinct and interpretable patterns of achievement completion that uniquely characterize each strategy, clearly emphasizing the specific tasks each was designed to target. Anecdotally, when provided with this table with blanked-out column names, an o1 model can accurately infer which achievement completion profiles correspond to each strategy, highlighting the clarity and effectiveness of policy steering.
> > >
> > > Collectively, these additional results unequivocally demonstrate that our Strategist framework generalizes effectively, delivering substantial improvements in exploration efficiency and task-specific performance across diverse scenarios. We hope these new results further strengthen the evidence for the broad applicability and effectiveness of our approach.
> > >
> > > Please let us know if there is anything else you would like us to clarify further or any remaining issues you wish us to address. We sincerely welcome your feedback to ensure the final version meets the highest possible standards.
> > >
> > > Thanks again for your constructive input and for helping us strengthen the paper.

---

### Official Review · Reviewer_vtAw · 2025-03-15

**Overall Recommendation:** 4

**Summary:**

This paper defines a new hierarchical RL framework through introducing a Strategy Problem: finding distributions over policies that balance specificity and value. This involves using LLM to generate sub-goals, and a reward shaping method to translate these sub-goals to quantitative feedback for RL. The proposed method aims to define a new type of a model-based RL, which uses LLM for sub-goal definition.

## update after rebuttal
I appreciate the authors responses. The authors have addressed my concerns, and I have no objections to accepting this paper.
I have increased my rating by a point.

**Claims And Evidence:**

We formalize strategy as a distribution over policies that balances specificity (pruning the search space) and
value (encompassing optimal plans).

Supported

We instantiate these ideas in the Strategist agent, which uses LLM-based tree search to encode domain knowledge into actionable top-down strategies without prespecifying their components. Crucially, we introduce a reward shaping methodology that translates strategies
into quantitative feedback for RL.

Supported

We empirically validate that this procedure enables effective exploration, leading to faster convergence than a classical PPO baseline.

Supported

**Essential References Not Discussed:**

If the authors would like to ground their method in human planning, they would have to discuss that literature, but I suspect that is not the goal.

One suggestion of a related paper: https://arxiv.org/pdf/2310.11614

**Experimental Designs Or Analyses:**

The authors describe their approach as human-like, however they neither discuss human behavioural literature, nor ground their framework in human data or human experiments. This claim should be removed.

The actual goal of this paper comes across as producing a model-based RL method that leverages LLM. I would still recommend accepting the paper based on this.

**Methods And Evaluation Criteria:**

A standard MDP formulation is defined. This is followed by a formal definition of specificity (a small set of policies) and value (there should exist in this small set a near-optimal policy). Then, Strategy problem is defines as finding a small set of policies that includes the near-optimal policy.

The authors describe a Strategist Framework that generates and explores potential strategies expressed in natural language. The framework is evaluated on a modified version of the Crafter environment, where the task is collecting meat at different levels of
difficulty.

(more details coming here...)

**Other Comments Or Suggestions:**

PPO in the abstract not defined

Line 171: PAC not defined

**Other Strengths And Weaknesses:**

Strength: Well written paper, lots of stuff in the Appendix. Has proofs and theory. I like the approach to formalizing what it means to search over a small number of sensible policies.

The main weakness is presentation: the authors are overselling their model as a New Paradigm in RL, which it is not. The sub-goal discovery method in this paper is new, and the authors can claim that they improve on existing approaches, but model-based RL is a very well established area.  The authors also claim human-likeness without a justification.

**Questions For Authors:**

.

**Relation To Broader Scientific Literature:**

The references related to RL models with LLM planning are well discussed, however the authors claim without evidence  their model to be human-like. There are many models of human planning, including models of human planning in problems that require simplification, or structural inference, reductive cognitive map representations .. they would need to be discussed if the authors wanted to justify their human-likeness.

**Theoretical Claims:**

I have read the paper, but have not carefully checked the proofs in the Appendix.

---

> ### Author Rebuttal · Authors · 2025-04-01
>
> We thank you for recognizing the strengths of our paper, including the clarity of writing and insightfulness of the formalism.
>
> ## Clarifying the "human-inspired" framing
> We acknowledge your concern regarding our use of "human-like" to describe the approach. Upon reflection, we agree that the term could imply unintended commitments to psychological realism, which was not our goal. We have revised the paper to **replace "human-like" with "human-inspired,"** which better captures our intent to indicate that our method leverages high-level strategic decomposition reminiscent of human planning processes, rather than modeling human cognition per se.
>
> While our primary objective indeed is not to ground our approach explicitly in human behavioral data or cognitive models, we recognize that some readers may find connections to this literature insightful. Therefore, we have included a **targeted discussion in the appendix**, briefly summarized here:
> > Hierarchical decomposition of goals has been central to human planning since the foundational work of Miller, Galanter, and Pribram (1960), and Lashley (1951). Humans naturally generate and evaluate multiple strategies, selecting among them by optimizing a cost–benefit tradeoff, as described by Lieder and Griffiths (2017). Moreover, humans utilize structural inference to abstract and simplify complex tasks (Gershman & Niv, 2010), supported by cognitive maps and mental models for flexible, model-based planning (Wang & Hayden, 2020). Lastly, the principle of bounded rationality highlights that humans regularly prune decision trees and favor simpler, cognitively less demanding plans (Huys et al., 2012; Lai & Gershman, 2024). These insights substantiate our strategist agent as balancing specificity (simplification and structure) against value (optimality), mirroring established principles of human decision-making.
>
> We hope this clarification addresses your concern effectively.
>
> ## Additional comments and revisions
> We appreciate the point regarding our characterization as a "New Paradigm in RL." Upon reconsideration, we agree this phrase could be overstated. We have thus **updated the description to "framework,"** which we believe is well supported by the formalization presented in Section 2.
>
> We also thank the reviewer for highlighting specific presentation issues, such as the undefined terms "PPO" and "PAC." We have addressed these explicitly in the revised version to enhance clarity.
>
> Finally, we have **expanded our experimental evaluation** to include **strong RL baselines (DreamerV3, EDE)** and **LLM-based approaches (LLM-as-policy, Eureka)**. For detailed results showing our method's advantages over these baselines, please see our responses to the other Reviewers.
>
> Thank you again for your valuable feedback.

---

### Decision · Program_Chairs · 2025-05-01

**Decision:**

Accept (poster)

**Comment:**

This paper introduces a top-down RL method called Strategist, which uses LLMs to generate high-level strategies and transform them into reward signals to learn policy. Experiments on a modified Crafter environment show that Strategist allows faster training convergence compared to standard RL algorithms.

The paper is well-written and clearly presents its motivation. The proposed integration of LLMs with a top-down RL approach is novel. The main weakness is its limited experiments, which are only verified in a single environment. While the results are promising, it remains unclear whether the method can generalize to other tasks.